# Satellite-derived emissions of carbon monoxide, ammonia, and nitrogen dioxide from the 2016 Horse River wildfire in the Fort McMurray area

Cristen Adams[1], Chris A. McLinden[2], Mark W. Shephard[2], Nolan Dickson[2], Enrico Dammers[2], Jack Chen[2], Paul Makar[2], Karen E. Cady-Pereira[3], Naomi Tam[1], Shailesh K. Kharol[2], Lok N. Lamsal[4,5], and Nickolay A. Krotkov[5]

[1]Environmental Monitoring and Science Division, Government of Alberta, Edmonton, Alberta, Canada
[2]Air Quality Research Division, Environment and Climate Change Canada, Downsview, Ontario, Canada
[3]Atmospheric and Environmental Research, Lexington, MA, USA
[4]Goddard Earth Sciences Technology and Research, Universities Space Research Association, Columbia, MD, USA
[5]Atmospheric Chemistry and Dynamic Laboratory, NASA Goddard Space Flight Center, Greenbelt, MD, USA

*Correspondence to*: Cristen Adams (cristen.adams@gov.ab.ca)

## Abstract

In May 2016, the Horse River wildfire led to the evacuation of ~88,000 people from Fort McMurray and surrounding areas and consumed ~590,000 ha of land in Northern Alberta and Saskatchewan. Within the plume, satellite instruments measured elevated values of CO, $NH_3$ and $NO_2$: CO was measured by two Infrared Atmospheric Sounding Interferometers (IASI-A and IASI-B), $NH_3$ by IASI-A, IASI-B and the Cross-track Infrared Sounder (CrIS), and $NO_2$ by the Ozone Monitoring Instrument (OMI). Daily emissions rates were calculated from the satellite measurements using fire hotspot information from the Moderate Resolution Imaging Spectroradiometer (MODIS) and wind information from the European Centre for Medium-Range Weather Forecasts (ECMWF) ERA5 reanalysis, combined with assumptions on lifetimes and the altitude range of the plume. Sensitivity tests were performed and it was found that uncertainties of emission estimates are more sensitive to the plume shape for CO, and to the lifetime for $NH_3$ and $NO_x$. The satellite-derived emissions rates were ~50-300 kt/d for CO, ~1-7 kt/d for $NH_3$, ~0.5-2 kt/d for $NO_x$ (expressed as NO) during the most active fire periods. The daily satellite-derived emissions estimates were found to correlate fairly well (R ~ 0.4-0.7) with daily output from the ECMWF Global Fire Assimilation System (GFAS) and the Environment and Climate Change Canada (ECCC) FireWork models, with agreement within a factor of two for most comparisons. Emission ratios of $NH_3$/CO, $NO_x$/CO, and $NO_x$/$NH_3$ were calculated and compared against enhancement ratios of surface concentrations measured at permanent surface air monitoring stations and by the Alberta Environment and Parks Mobile Air Monitoring Laboratory (MAML). For $NH_3$/CO, the satellite emission ratios of ~0.02 are within a factor of two of the model emission ratios and surface enhancement ratios. For $NO_x$/CO satellite-measured emission ratios of ~0.01 are lower than the modelled emission ratios of 0.033 for GFAS and 0.014 for FireWork, but are larger than the surface enhancement ratios of ~0.003, which may have been affected by the short lifetime of $NO_x$. Total emissions from the Horse River fire for May 2016 were calculated and compared against total annual anthropogenic emissions

for the province of Alberta in 2016 from the ECCC Air Pollutant Emissions Inventory (APEI). Satellite-measured emissions of CO are ~1500 kt for the Horse River fire and exceed the total annual Alberta anthropogenic CO emissions of 992.6 kt for 2016. The satellite-measured emissions during the Horse River fire of ~30 kt of $NH_3$ and ~7 kt of $NO_x$ (expressed as NO), are approximately 20% and 1% of the magnitude of total annual Alberta anthropogenic emissions, respectively.

## 1 Introduction

The 2016 Horse River wildfire (MNP LLP, 2017) was first observed on 1 May 2016 approximately 7 km southwest of Fort McMurray and grew rapidly. On 3 May 2016, the fire entered the urban service area of Fort McMurray, leading to the evacuation of ~88,000 people, damaging multiple structures, and threatening nearby communities, oil sands camps and

10 facilities. The fire then moved eastward toward Saskatchewan and, after two months of fire-fighting efforts, was determined under control on 4 July 2016. The fire consumed 589,522 ha in Northern Alberta and Saskatchewan and led to an estimated 3.58 billion dollars in insured losses (Insurance Bureau of Canada, 2016).

The number and size of wildfires in North America have been increasing over the past few decades due to various factors, 15 such as drought conditions, changes in land-use, and fire management practices (Romero-Lankao et al., 2014). In Canada the number of wildfires and total burned area have increased between 1961 and 2015 (Landis et al., 2018). In Alberta, the number of wildfires has increased, but the total annual acreage burned has been variable, with several years of anomalously large burn areas (Landis et al., 2018). The annual burn area in the boreal forest is projected to increase, with some of the largest increases predicted in north-eastern Alberta (Boulanger et al., 2014), which could affect ecosystems (e.g., Stralberg et al., 2018), air 20 quality (e.g., Yue et al., 2015), and the climate (e.g., Oris et al., 2014).

Emissions from forest fires have a large effect on global and regional air quality; more than 10% of global CO from wildfires is emitted over mid- and high-latitude (Langmann et al., 2009). Emission factors characterize the amount of a gas released per kg of fuel consumed (g/kg) and are often used in air quality models to estimate emissions from wildfires. Emission factors for 25 a given land-cover type vary naturally depending on burning conditions (flaming versus smouldering phases) and fuel type (stems, leaves or needles). The flaming versus smouldering phases of combustion also have different emission factors for the same species. For example, $NO_x$ and $CO_2$ are emitted in larger quantities during flaming periods, while $NH_3$ and CO are emitted in larger quantities during smouldering periods (e.g., Andreae and Merlet, 2001). The occurrence of flaming and smouldering phases depend on the moisture content of the fuel (Chen et al., 2010) and can vary diurnally, with more active 30 flaming during the day, as well as spatially between the fire front and areas of residual burning (e.g., Langmann et al., 2009; McRae et al., 2005). Emission factors also vary with fuel type for, e.g., stems, leaves, or needles. A study in northwestern Alberta found that deciduous trees (e.g., trembling aspen) contain more nitrogen than conifers (e.g., white spruce) (Jerabkova

et al., 2006); this might be expected to lead to larger emissions of $NO_x$ and $NH_3$ for trembling aspen. Akagi et al. (2011) estimated variability in boreal forest emission factors from aircraft and laboratory studies of 35% for CO, 85% for $NH_3$, and 77% for $NO_x$.

Due to their spatial and temporal coverage, satellite remote sensing instruments can capture some of this natural variability, by estimating emissions and emission factors over various seasons, burning conditions, fuel types, and moisture content. Previous studies using satellite CO, $NH_3$, and $NO_2$ data have assessed the relationship between these species (e.g., Coheur et al., 2009; Krol et al., 2013; Luo et al., 2015; Paulot et al., 2017; Pechony et al., 2013; R'Honi et al., 2013; Whitburn et al., 2015; Yurganov et al., 2011), and have also estimated emissions and emission factors (e.g., Mebust and Cohen, 2014; Mebust

et al., 2011; Schreier et al., 2014, 2015; Tanimoto et al., 2015; Whitburn et al., 2015, 2016b). Emission factors have also been estimated using ground-based Fourier Transform Infra-Red spectrometers (e.g., Lutsch et al., 2016).

The Horse River fire provides an excellent opportunity to evaluate methods for estimating emissions from satellite. Due to the close proximity of the fire to industry and communities, there is a relatively extensive surface monitoring network in the

region, and many surface air quality datasets are collected routinely in the area. These networks captured detailed information about the wildfire plume (Landis et al., 2018; Wentworth et al., 2018), which can be used to support the satellite-based estimates. In this study, we estimate emissions from several satellite instruments: CO measured by the two Infrared Atmospheric Sounding Interferometers (IASI-A and IASI-B), $NH_3$ measured by IASI-A, IASI-B and the Cross-track Infrared Sounder (CrIS), and $NO_2$ measured by the Ozone Monitoring Instrument (OMI). We compare emissions estimates with model

data, explore relationships between CO, $NH_3$ and $NO_x$, and we derive emission factors from the satellite data.

## 2    Datasets

Satellite datasets of CO, $NH_3$, and $NO_2$ vertical column densities (VCDs) were used along with Moderate Resolution Imaging Spectroradiometer (MODIS) fire radiative power (FRP) data to estimate emissions. The emissions estimates were compared

against two model datasets. These datasets are described in the subsections below.

### 2.1    IASI CO and NH₃

In this study, we used the most recent CO and $NH_3$ data products, which can be found at http://iasi.aeris-data.fr/. The observations are made with the Infrared Atmospheric Sounding Interferometer (IASI-A and –B)  instruments on board of the

MetOp satellites. The infrared sounders have a sun-synchronous orbit and cover the globe twice a day with overpasses at 9:30

and 21:30 local solar time (LST). The instruments have a swath width of about 2,200 km, with an individual pixel footprint size of 12 km diameter at nadir, which increases in size towards the sides of the swath (Clerbaux et al., 2009).

For CO we use the FORLI-CO product (Hurtmans et al., 2012). The IASI-CO product has been validated with aircraft
observations which showed overall good results and performance (George et al., 2015). IASI CO VCDs have also been compared against ground-based Fourier transform infrared (FTIR) spectrometer measurements, with typical differences of ~10% when FTIR profiles are smoothed by the IASI averaging kernels (Kerzenmacher et al., 2012). During the fire, total column averaging kernels showed increased sensitivity at surface with values of ~0.4-0.6 CO for large VCDs greater than $3.5 \times 10^{18}$ molec/cm$^2$ (not shown here). These large VCD measurements are taken within the smoke plume, and are the primary
contributor to the emissions estimates. This is consistent with previous studies which have found increased sensitivity to surface CO for large VCDs (Bauduin et al., 2017). Yurganov et al. (2011) compared IASI VCDs with measurements from three grating spectrometers within a plume from forest and peat fires in Central Russia in July-August 2010. They found that the IASI VCDs were biased low compared with the ground-based measurements by an estimate of $1.61 \times 10^{18}$ molec/cm$^2$ or ~35%, over a sample with a mean IASI CO VCD of $4.7 \times 10^{18}$ molec/cm$^2$.

For NH$_3$, we use the current IASI-NNv2.1 product by Van Damme et al. (2017). The product combines the calculation of the dimensionless spectral index (HRI) with a neural network to calculate the total columns of NH$_3$ (Whitburn et al., 2016a). The neural network uses a set of parameters to find the most representative state of the atmosphere. The previous versions of the NH$_3$ product were validated by FTIR observations (Dammers et al., 2015, 2017) and were found to underestimate the total
columns by 40%, with an increase at low local concentrations and a better performance for regions with higher local concentrations. Only observations with a cloud cover below 25% were used in this study. Averaging kernels are not produced as a part of the NH$_3$ retrievals; however, previous studies have demonstrated good agreement with surface and FTIR measurements (e.g., Clarisse et al., 2010; Van Damme et al., 2015; Dammers et al., 2017), demonstrating that there is sensitivity to the lower layers of the atmosphere.

## 2.2   CrIS NH$_3$

The CrIS instrument is flown on board the Suomi National Polar-orbiting Partnership (S-NPP) satellite with an overpass of ~1:30 and 13:30 LST and a pixel spatial resolution of 14-km circles at nadir. The CrIS Fast Physical Retrieval (CFPR) uses an optimal estimation method (Rodgers, 2000) to produce vertical profiles of NH$_3$ and associated error estimates and vertical
sensitivity (averaging kernels) (Shephard and Cady-Pereira, 2015). CFPR NH$_3$ retrievals achieve valid single pixel retrievals down to the 0.2 to 0.3 ppbv range (Kharol et al., 2017). In this study, these retrieved profiles were then integrated to compute VCDs. Initial CrIS validation studies against ground-based FTIR VCDs show a correlation of 0.77, and very little bias (+2%) (Dammers et al., 2017). CrIS total column averaging kernels during the fire suggest good sensitivity to NH$_3$ in lower layers

of the atmosphere, with values of ~0.5-1.5 for the 0-3 km altitude range (not shown here). Above ~3 km, the sensitivity is very low, which is expected if ammonia concentrations are low at these altitudes (see Sect. 4.2). For the present study, CrIS data with quality flag = 4 were included.

**2.3 OMI NO$_2$**

$NO_2$ VCDs were obtained from the v3 Standard Product (SP) (Krotkov et al., 2017) of the Ozone Monitoring Instrument (OMI) (Levelt et al., 2006), which measures UV-visible sunlight in the nadir viewing geometry with a 13 km x 24 km footprint. An important component of any UV-visible $NO_2$ algorithm is the air mass factors (AMF), which accounts for the sensitivity of the sensor to $NO_2$ and is based on several factors including the shape of the $NO_2$ profile, aerosols, and surface reflectivity.
Here, the SP AMFs were replaced with the Environment and Climate Change Canada (ECCC) AMFs because they are optimized for the measurement area (McLinden et al., 2014).

Smoke affects the scattering of UV-visible sunlight, and therefore affects the OMI $NO_2$ AMFs, and ultimately the VCD retrievals. The presence of large smoke plumes also complicated the determination of cloud fraction. OMI provides a measure of the cloud radiative fraction (CRF); observations with CRF values exceeding a certain threshold (typically 0.5) are generally
excluded from analysis, since cloud prevents OMI from detecting the $NO_2$ below. Daily MODIS true colour maps were inspected for each day in May. The true color maps visually demonstrated that several days were heavily contaminated by clouds, and these were omitted from the study. Several other days, however, appeared cloud-free despite a large reported CRF in pixels that contained smoke plumes. This suggests that the smoke itself was being identified as cloud. As a result, on these
days, the CRF was set to zero for large $NO_2$ VCDs ($> 1x10^{15}$ molec/cm$^2$) and the OMI AMFs were modified to account for the effect of the smoke plume on the $NO_2$ VCDs (see Sect. 4.4).

Furthermore, to improve sampling over fire hot spots, some data affected by the row anomaly were also included. The row anomaly causes changes in radiance signal measurements collected by specific OMI pixels starting in 2008 (Torres et al.,
2018) and are not included in the v3 dataset. For larger VCDs, such as those that occurred during the wildfire, the uncertainty due to the row anomaly is less significant, particularly when considering other sources of uncertainty in the emissions estimates. Furthermore, the de-striping algorithm is able to remove much of the bias associated with the row anomaly. Therefore, in order to increase sampling near the fire hot spots, data affected by the row anomaly from the v2 OMI $NO_2$ standard product (Bucsela et al., 2013) were included in the analysis for VCDs greater than $1x10^{15}$ molec/cm$^2$. The differences
between v2 and v3 $NO_2$ have a large impact in the stratosphere, but very little impact in the troposphere (van Geffen et al., 2015; Marchenko et al., 2015). Inclusion of the row anomaly data was required to have sufficient sampling to estimate emissions for 6 May, 13 May, 15 May, and 24 May. The row anomaly did not affect VCDs over the fire hot spots for other days.

## 2.4    MODIS fire radiative power and hotspots

The MODIS instruments, on board the National Aeronautics and Space Administration (NASA) Earth Observation System Terra and Aqua satellites, detect fires using data collected in the infrared and spectral channels (Kaufman et al., 1998). The MODIS Active Fire Product MCD14ML FRP data product (Giglio et al., 2003, 2006) was used to identify and quantify fire hotspots. The MODIS fire product is publicly available at: http://modis-fire.umd.edu/index.php. Daytime measurements from the Terra 10:30 LST descending node and the Aqua 13:30 LST descending nodes were used in this study. Note that MODIS does not detect all fires as clouds can obscure hot spots.

## 2.5    GFAS model

The Global Fire Assimilation System v1.2 (GFAS) (Kaiser et al., 2012) uses assimilated FRP from the MODIS instruments in order to estimate daily emissions from biomass burning on a 0.5°×0.5° global grid. GFAS converts the assimilated FRP to combustion rates using conversion factors, which are prescribed for different land cover types based on regressions between GFAS FRP and dry matter combustion rates from the Global Fire Emissions Database (GFED) model. Emission factors are applied to the combustion rates to estimate emissions for 40 gas-phase and aerosol species. Emission factors are obtained from Andreae and Merlet (2001) and combined with updates from more recent literature. GFAS data are publicly available at http://apps.ecmwf.int/datasets/data/cams-gfas/.

## 2.6    FireWork Model

The ECCC FireWork model (Pavlovic et al., 2016) is Canada's national operational air quality forecast model with near-real-time biomass burning emissions. The model estimates fire emissions via the bottom-up approach by combining near-real-time fire location with pre-defined fire size, and estimated fuel consumption information from the Canada Forest Service's Canada Wildland Fire Information System (CWFIS), with emission factors based on US Forest Service's Fire Emissions Prediction System (FEPS). Fire hotspot information during May 2016 were assembled from three satellite sensors: the Advanced Very High Resolution Radiometer (AVHRR), MODIS, and the Visible Infrared Imaging Radiometer Suite (VIIRS), acquired through the National Oceanic and Atmospheric Administration (NOAA), NASA, University of Maryland, and US Forest Service's Remote Sensing Applications Center (RSAC). Total fuel consumption at individual hotspots was calculated for meteorology at noon LST. Modelled daily fire emissions were temporally distributed to hourly intervals using a fixed diurnal profile. The model setup assumes fire persistence where past 24-hour fire hotspots are used to calculate future 48-hour emissions for the forecasting system.

## 2.7 Surface air monitoring

The satellite-derived and model emissions estimates were complemented by surface air monitoring data. Continuous surface measurements of CO, $NH_3$, and $NO_x$ were collected as a part of monitoring by the Wood Buffalo Environmental Association at two permanent air monitoring stations located in Fort McMurray: Fort McMurray-Patricia McInnes station (56.751378°N, 111.476694°W) and Fort McMurray-Athabasca Valley station (56.733392°N, 111.390501°W). The permanent air monitoring station data were obtained from the Alberta Airdata Warehouse http://airdata.alberta.ca/ retrieved on 20 July 2017. CO was measured at the Fort McMurray – Athabasca Valley station using a ThermoScientific Model 48i gas filter correlation analyser. $NH_3$ was measured at the Fort McMurray – Patricia McInnes station using a Teledyne API Model T201 chemiluminescence analyser. $NO_x$ was measured at both Fort McMurray stations using Model T200 chemiluminescence analysers. The data presented in this study are hourly averages. The data collected at these stations, as well as other stations in the area during the Horse River fire, have been analysed and reported on by Landis et al. (2018).

Additional continuous measurements of CO, $NH_3$, and $NO_x$ surface concentration were also collected by the Alberta Environment and Parks Mobile Air Monitoring Laboratory (MAML), which was deployed in response to the wildfire and took measurements at various locations within Fort McMurray to support emergency response personnel and re-entry to the community between 16 May and 9 June 2016. The MAML is a 27-foot (8.2 m) vehicle which is equipped with air monitoring instruments. For this study, we consider CO measured Thermo 48i-TLE (trace-level) continuous gas analyzer and $NH_3$/$NO_x$ by a Thermo 17i continuous gas analyzer. Hourly average measurements from the MAML collected during the wildfire and the locations sampled are provided in the supplementary material. Hourly averages were calculated from the data, requiring 45 minutes of measurements for a valid hourly average.

## 3 Formulae for Emissions Estimates

The methodology used to calculate satellite emissions is based on the work of Mebust et al (2011), though the practical implementation of the formulae is different, as described in Sect. 4. In this section, the formulae used to calculate emissions estimates are presented for the general case.

Consider a Gaussian model for a plume travelling downwind in the $x$ direction. The concentration, $C$ (molec/cm$^3$), at a point downwind is given by

$$C(x, y, z) = \frac{E}{2\pi u \sigma_y \sigma_z} \cdot \exp\left[-\left(\frac{y}{\sqrt{2}\sigma_y}\right)^2\right] \cdot \exp\left[-\left(\frac{z}{\sqrt{2}\sigma_z}\right)^2\right] \cdot e^{-x/\tau u} \quad , \tag{1}$$

where $\tau$ is the lifetime (s), $u$ is windspeed (m/s), $E$ is the rate of emissions (molecs/s), and $\sigma_y$ and $\sigma_z$ are constants related to diffusion in the crosswind direction (y) and in altitude (z), respectively (Stockie, 2011). This is a solution of the steady-state

advection diffusion equation for constant diffusion coefficients, no deposition, and constant wind, assuming diffusion can be neglected in the downwind direction.

This can be converted into a VCD (molec/cm$^2$) by integrating over altitude, $z$,

$$V(x, y) = \frac{E}{\sqrt{2\pi}u\sigma_y(x)} \cdot \exp\left[-\left(\frac{y}{\sqrt{2}\sigma_y(x)}\right)^2\right] \cdot e^{-x/\tau u} \ . \tag{2}$$

Integrating in the cross-wind direction, $y$, then one obtains a line density, L (molec/cm) as a function of distance downwind

$$L(x) = \frac{E}{u} \cdot e^{-x/\tau u} \tag{3}$$

While the Gaussian plume model was a convenient starting point, Eq. 3 can be seen to satisfy the 1D advection-diffusion equation directly (e.g., Stockie, 2011). Thus, Eq. 3 is valid in the steady-state assuming constant emissions, constant diffusion coefficients, wind, and chemical decay, and assuming diffusion can be neglected in the downwind direction. Equation 3 can be used to infer the lifetime of the species by fitting an exponential to $L(x)$ as a function of downwind distance $x$ and using an assumed value for $u$.

If we integrate $L(x)$ over a distance downwind, e.g., from 0 to $x_c$, then we get the mass (kg) from the source to the downwind location

$$m = \int L(x)\, dx = \frac{E}{u}\int_0^{x_c} e^{-x/(\tau \cdot u)} dx = \frac{E}{u} \cdot \tau \cdot u \cdot (1 - e^{-x_c/(\tau \cdot u)}) = E \cdot \tau \cdot (1 - e^{-\frac{t_c}{\tau}}), \tag{4}$$

where $t_c$ (s), the residence time inside the box, and $x_c$, the distance to the edge of the box, are related by windspeed, $u$.

Solving for emissions, $E$, we get

$$E = \frac{m}{\tau \cdot (1 - e^{-\frac{t_c}{\tau}})} \ . \tag{5}$$

Equation 5 is used to calculate emissions by Mebust et al. (2011) and in the present study. This assume flat homogeneous terrain and continuous wind patterns.

25 **4    Calculation of Emissions from Satellite Measurements**

For each day, the satellite data were averaged onto a 4x4 km$^2$ grid which was oriented so that emissions originate at the centre of the grid and propagate downwind in the $+x$ direction. The centre of the grid was set to the mean latitude and longitude of the fire hotspots, weighted by MODIS FRP for the given day. For IASI-morning measurements, MODIS-Terra FRP data were used because the overpass times were similar. For the IASI-evening, CrIS, and OMI measurements, the MODIS-Aqua FRP

data were used. In order to orient the grid, the mean wind direction at the hot spots was calculated from European Centre for Medium-Range Weather Forecasts (ECMWF) ERA5 (https://software.ecmwf.int/wiki/display/CKB/ERA5+data+documentation) assimilation wind direction profiles. The wind profiles were interpolated to the fire hotspot locations and MODIS overpass time, yielding a set of wind profiles for each fire

hot spot. The wind profiles were then weighted by an assumed vertical smoke profile (see Sect. 4.2) and averaged. The winds at each fire hot spot were then weighted by FRP and averaged over the fire hotspots to yield a single wind direction for the fires. The winds were weighted by FRP because larger FRP is associated with more active burning and, therefore, a larger contribution to emissions.

Figure 1 shows the original sampling of the satellite data and the smoothed, rotated data for 16 May 2016. Satellite data for the given day were oversampled onto the 4x4 $km^2$ grid described above. For each VCD measurement, all grid-points that were within the satellite pixel were assigned to that VCD value. This yielded vectors of oversampled latitude, longitude, VCD, and weights. These data were then averaged onto the 4x4 $km^2$ grid using an averaging radius of 15 km, and weighting the measurements by the inverse of the pixel area. Days with large gaps in gridded VCDs in the area of the fire hotspots and

downwind of the fires were excluded from the analysis. This was done by visually inspecting the original and gridded VCDs for each day; if gaps in the data covered large areas that were required to resolve the plume or led to interpolation of the plume that looked suspect, the day's data was excluded. Most of the days excluded were missing more than half of the data over or downwind of the fire hot spots. For days with sufficient data, small gaps in the gridded dataset were filled, using interpolation with the inpaint_nans function in MATLAB (D'Errico, 2009).

The baseline VCDs were subtracted from the daily VCDs, yielding dVCD, the portion of the VCD from fire emissions. Baseline VCDs were calculated using data from May 2014, a year/month that was not strongly affected by forest fire smoke and the data were filtered to remove peak values (> 99 percentile). For each day in 2016, background was estimated using the filtered data for the entire month of May 2014, which was oversampled, gridded and rotated using the winds for the day in

2016, as described above. This was done so that baseline values which varied spatially, due to e.g., emissions from the oil sands, were accounted for. For CO and $NH_3$, the baseline VCDs did not have a strong systematic variation over the measurement area, as demonstrated by the mean and 1-σ of the May 2014 VCDs, which were filtered to remove peak values (> 99 percentile). For CO, mean and 1-σ baseline values had small variability and were consistent across IASI instruments, with values of $(2.0\pm0.2)\times10^{18}$ molec/$cm^2$ for IASI-A morning, $(1.9\pm0.2)\times10^{18}$ molec/$cm^2$ for IASI-A evening, $(2.0\pm0.2)\times10^{18}$

molec/$cm^2$ for IASI-B morning, and $(1.9\pm0.2)\times10^{18}$ molec/$cm^2$ for IASI-B evening. The CO baseline VCDs are significant compared with the enhanced levels of CO within the plume ~3-5$\times10^{18}$ molec/$cm^2$. For IASI $NH_3$, background values were $(0.6\pm3.7)\times10^{15}$ molec/$cm^2$ for IASI-A morning, $(1.1\pm0.9)\times10^{15}$ molec/$cm^2$ for IASI-A evening, $(-0.8\pm3.3)\times10^{15}$ molec/$cm^2$ for IASI-B morning, and $(-1.1\pm9.3)\times10^{15}$ molec/$cm^2$ for IASI-B evening. These values have more variability compared with CO and show biases between IASI instruments, but they are small (~5%) compared with the enhanced VCDs observed over the

smoke plume of ~5-15×10$^{16}$ molec/cm$^2$ and therefore any systematic differences will have a small influence on the emissions estimates. The baseline VCDs for CrIS are larger than for IASI, with values of $(1.2\pm1.7)\times10^{16}$ for day and $(0.9\pm1.6)\times10^{16}$ for night. The differences between baseline VCDs for CrIS and IASI are due to differences in the retrieval methods, which affect results when VCDs approach zero. The baseline VCDs for NO$_2$ reflect local emissions sources and ranged from $3\times10^{14}$ molec/cm$^2$ over remote regions to $2\times10^{15}$ molec/cm$^2$ over the mineable oil sands region. The NO$_2$ baseline values over the oil sands are on the same order of magnitude as the enhanced levels of NO$_2$ in the plume ~5-15x10$^{15}$ molec/cm$^2$. The 2014 baseline VCDs for NO$_2$ were filtered for CRF < 0.3 and did not include any data affected by the row anomaly.

Next, boxes were drawn around dVCD, as shown for 16 May 2016 in Figure 1. The width of the box in the cross-wind direction, $y$, was +/- 150 km from the centre of the fire hotspots. This was large enough to encompass the plume up to 200 km downwind of the fire, as determined by examining VCD maps for each day. Upwind, $-x$, the top of the box was set to the first grid value to encompass all of the fire hotspots. Downwind, $+x$, the box was drawn 20 km downwind of the most downwind hotspot to ensure that all of the hotspots were enclosed in the box, when accounting for the satellite footprints.

Within the box, emissions were calculated using Eq. 5. The mass, $m$, of CO, NH$_3$ or NO$_2$ in the box, was calculated by integrating gridded dVCDs within the box and multiplying by the molecular mass of the species. $t_c$ was calculated from $u$, the mean downwind wind speed, and $x_c$ the distance to the centre of the fires (or the centre of the grid). The mean downwind wind speed was calculated using ECMWF ERA5 winds at the pixels of the satellite measurements within the box. The wind profiles were interpolated to the satellite measurement locations and the weighted average of winds over the vertical profile of the assumed plume shape was calculated. At each satellite pixel location, the component of the wind speed in the downwind direction was taken. Then the average of the downwind wind speed components was calculated. $\tau$ was calculated from the data by examining the decay curves of NO$_2$ and NH$_3$ downwind of the fires, as described in Sect. 4.3.

The uncertainties associated with the emission estimates were calculated from a series of sensitivity tests. These tests were intended to determine the sensitivity of the emissions to different choices, which we have labelled as analysis settings, for the method, the plume shape, species lifetimes, handling the effect of smoke, the conversion of NO$_2$ to NO$_x$ and diurnal variation of emissions. The calculated sensitivities provide at least a rough estimate of the uncertainty in the emission estimates. The subsections below describe how each analysis setting was chosen and how associated sensitivity tests were performed. In order to run the sensitivity tests described, emissions estimates were recalculated using the alternate settings for days with sufficiently large emission estimates for the default settings (> 50 kt for CO; > 1 kt for NH$_3$; > 0.5 kt for NO$_x$). The estimated uncertainty was the mean percent difference in emissions between the sensitivity test and the default settings across all days. In addition to the sensitivity tests, the uncertainty in VCDs due to satellite retrievals was also accounted for. Table 1 summarizes the analysis settings and associated uncertainty for CO, NH$_3$, and NO$_x$ emissions estimates. The total uncertainty

in emissions estimates, calculated by adding the various uncertainty terms in quadrature, was 67% for IASI CO, 82% for IASI NH$_3$, 74% for CrIS NH$_3$, and 73% for OMI NO$_x$.

## 4.1    Method Uncertainty

In order to estimate the uncertainty due to the selected emissions estimate method, an alternate method was tested using IASI CO, based on one-dimensional fluxes downwind of the fire. For the long lifetime of CO, Eq. 3 reduces to $E = L(x) \cdot u$. The downwind flux emission estimate was calculated 4 km downwind of the fire, using the line density $L(x)$ and the downwind wind-speed components, and yielded emissions estimates that were on average 48.8% lower than the rectangular integration method. Similar values were obtained for one-dimensional flux calculations at 20 km downwind. The rectangular integration

method (Eq. 5) was used in this study instead of the downwind flux method because rectangular integration includes the highest measured values of CO, NH$_3$, and NO$_2$ right over the fire hotspots. Furthermore, the rectangular integration method is less sensitive to the assumed lifetime of NO$_2$ and NH$_3$ than the one-dimensional flux method. Note that the size of the rectangle used for the rectangular integration method was also varied to test method uncertainty, but had a relatively small influence (~5%) on the emissions estimates. The value of 48.8% method uncertainty was used for both NH$_3$ and NO$_x$ because the

alternate method used in the sensitivity tests is based on downwind flux, where NH$_3$ and NO$_x$ line densities are smaller due to their short life-times. Therefore, the alternate method is very sensitive to the assumed lifetimes and is therefore not appropriate for these species.

## 4.2    Plume Profile Shape

The smoke profile shape is used to weight the ECWMF wind profiles used in the flux calculations. For example, if the plume is at the surface, surface winds will dominate the flux calculations. If the plume is aloft 2-4 km, then winds at 2-4 km, which are typically larger than the surface winds will be more important. Therefore, higher wind-speeds lead to larger emissions estimates through Eq. 5, where $t_c = x_o/u$. In order to infer the shape of the plume profile, several datasets were used, including surface monitoring, satellite, and models, as described in the paragraphs below.

Surface air monitoring stations in the Fort McMurray area were affected by the plume and recorded enhanced levels of many parameters, including PM$_{2.5}$, SO$_2$, NO$_x$, black carbon, NH$_3$, and CH$_4$ at locations both very near the fire and downwind (Landis et al., 2018). This suggests that the smoke profile reached the surface and was not strictly aloft. Therefore, all plume profile shapes tested extended to the surface.

On 5 May 2016, the wind directions varied with altitude and were compared against CrIS NH$_3$ VCDs to infer the plume altitude range, as shown in Figure 2. CrIS NH$_3$ was the only VCD dataset with sufficient sampling to resolve the downwind shape of

the plume on this day. The VCDs were gridded on a 4x4 km grid centred at the fire hotspots, which was not rotated. The ECMWF winds within a 40x40 km$^2$ box around the fires were averaged, yielding the wind directions shown in the figure. The direction of the NH$_3$ plume aligns best with winds between 1000-800 hPa (approximately 0-2 km), suggesting that the bulk of the NH$_3$ plume is within this altitude range.

Two additional satellite datasets and a model dataset were considered in order to estimate the altitude of the plume top. Smoke plume heights were retrieved using the NASA Multi-angle Imaging SpectroRadiometer (MISR) Interactive Explorer (MINX) software (Nelson et al., 2013). MINX uses imagery obtained from the MISR instrument's nine multi-angled cameras to perform the stereoscopic retrieval of heights and motion vectors for smoke plumes. A stereoscopic height retrieval algorithm
was developed to focus specifically on aerosol plumes at the highest spatial resolution possible with the MISR data. This algorithm relies on the manual determination of smoke plume locations, boundaries and wind directions. Nine distinct smoke plumes were visible by MISR over Fort McMurray during the month of May 2016, and were sampled over four days (4, 6, 15, and 18 May). All available pixels within the nine smoke plumes were binned over the distance downwind of the fire, as shown in Figure 3. MISR plume heights within 100 km downwind of the centre of the fire were ~3 km above sea level. Plume
heights were also measured by the Cloud-Aerosol Lidar and Infrared Pathfinder Satellite Observations (CALIPSO), which passed over the wildfire area on 14 and 15 May, using the vertical feature map (VFM) dataset (Winker et al., 2009). For measurements identified as "smoke", the mean plume height within 110-112°W and 55-59°N was 2.9±0.2 km (N = 18) on 14 May and 3.0±0.1 km (N = 81) on 15 May, where the error range denotes the standard deviation of the mean and N is the number of measurements in the average. Therefore, both the MISR and CALIPSO satellite datasets suggest that the plume
reached a maximum altitude of ~3 km. The GFAS model also calculates information on plume altitude using a plume rise model (Rémy et al., 2017). For the Horse River fire, daily average plume tops ranged from ~2-4.5 km, while the mean altitude of maximum injection ranged from ~1-3 km.

Given the uncertainties in the shape of the plume profile, several plume shapes were considered in sensitivity tests for this
analysis, as shown in Figure 4. A box-shaped plume for 0-3 km was selected as the default setting, since most of the datasets considered suggest that the plume was within this altitude range. In addition, a box-shaped plume extending from 0-6 km was selected to account for higher possible plume heights. The "NH$_3$ profile" plume shape was derived from the mean CrIS daytime NH$_3$ profile during the fire for VCDs greater than 8x10$^{16}$ molec/cm$^2$, and is weighted more heavily toward the surface. The NH$_3$ profile shape captures the possible contribution of residual smouldering to emissions, which are not lofted into the
plume and may stay near the surface. The triangle plume shape peaks at ~ 2 km and was calculated from climatological May smoke injection profiles for latitudes of 54-60°N and longitudes of 100-120°W (Sofiev et al., 2013), which are publicly available at http://is4fires.fmi.fi/. For all species, the largest difference in emissions estimates from the default plume were obtained when adopting the NH$_3$ profile plume shape, and therefore this shape was used to determine the uncertainties in Table 1. The emissions estimates for CO are more sensitive to the plume profile shape than the emissions estimates for NH$_3$ and

NO$_2$. This is because for long life-times $\tau \gg t_c$, Eq. 5 reduces to $E \sim m/t_c = m \cdot u/x_c$, and therefore the emissions estimate is directly proportional to the plume wind-speed, $u$.

### 4.3 Lifetimes of CO, NH₃, NO₂

The lifetimes of CO, NH$_3$, and NO$_x$ within wildfire plumes affect emissions estimates through Eq. 5. Previous satellite studies have assumed lifetimes of ~1-2 weeks for CO (e.g. Whitburn et al., 2015), which is much longer than the ~1-5 h timescale for the plume to clear the box used to estimate emissions in the present study. Therefore, a 2-week lifetime was used in the calculations, but the effect of varying this lifetime to, e.g., 1 week or infinity is negligible and not included in the error budget. For NH$_3$, previous satellite emissions estimate studies assume lifetimes of 12-36 h (Whitburn et al., 2015 and references therein) and Lutsch et al. (2016) estimated a lifetime of 48 h using a ground-based FTIR. However, those studies use methods which average over longer time periods and therefore consider NH$_3$ further downwind than in the present study. Closer to the emissions source, lifetimes could be shorter (Paulot et al., 2017). For example, aircraft observations in a smoke plume near a wildfire in the Alaska boreal forest found that of NH$_3$/CO decreased by $1/e$ in 2.5 h (Goode et al., 2000). Aircraft observations over Chapparel fires in California found that NH$_3$/CO decreased by a half in 4.5 h (Akagi et al., 2012), corresponding to a lifetime of ~6 h, but most of the decrease in NH$_3$/CO occurred in the first hour, suggesting rapid initial loss of NH$_3$. For satellite-derived NO$_x$ emissions estimates, Schreier et al. (2015) assumed a 6 h lifetime based on satellite measurements over megacities (Beirle et al., 2011). Mebust et al. (2014; 2011) used a shorter lifetime of 2 h, based on several observations of lifetimes in plumes ranging from 2-3 h. Tanimoto et al. (2015) calculate two alternate sets of emissions for NO$_2$, using both lifetimes of 2 h and 6 h.

For the present study, lifetimes of NH$_3$ and NO$_x$ were estimated directly from the satellite VCD datasets, as shown in Figure 5. For each day with sufficient data, the line density of NH$_3$ and NO$_2$, $L(x)$, as a function of downwind distance from the centre of the fire hotspots, $x$, was calculated. $L(x)$ was calculated using data within ±150 km cross-wind. Based on Eq. 3, an exponential was fitted to the $L(x)$ curve in order to calculate the lifetime $\tau$, as shown for example on 16 May. Then, using the exponential fit parameter and the weighted average wind-speed of the plume, $u$, the lifetime, $\tau$, was calculated. The fitted life-times are also shown in Figure 5, with errorbars denoting the 95% confidence intervals of the exponential fit parameter. Life-times are only shown for days with sufficiently large plumes (emissions > 1 kt/d for NH$_3$ or > 0.5 kt/d for NO$_x$) and small fitting errors (< 1 h for both NH$_3$ and NO$_2$). Approximately 60% of ammonia fits with sufficiently large plumes did not meet the fitting error criteria; this occurred primarily when wind speeds were low and/or winds were variable downwind, and therefore the plume changed direction and/or accumulated over some downwind locations. All NO$_2$ fits met the fitting criteria, as the short lifetime of NO$_2$ makes it less sensitive to inconsistencies in the winds far downwind. The fitted life-times were calculated using wind-speeds from both the 0-3 km box profile shape, as well as the alternate wind profile shapes shown in Figure 4. The plume weighted toward the surface (NH$_3$ profile) yielded longer life-times due to lower wind-speeds. Note that

the effect of other parameters included in the sensitivity tests, such as $NO_2$ AMFs were also tested but had a very small impact on the fitted life-times.

Based on the fitted lifetimes, values of $\tau = 3$ h for $NH_3$ and $\tau = 1.5$ hours for $NO_2$ were used in this analysis. The $NH_3$ lifetime of 3 h is much smaller than the other satellite emissions studies described above, but is reasonable given that the emissions estimate method considers $NH_3$ very near the emissions source, where lifetimes could be shorter. For the sensitivity tests, minimum values of 1.5 h for $NH_3$ and 1 h for $NO_2$ were derived from the lower range of the fitted lifetime values. For $NH_3$ and $NO_2$, upper range values of 12 h and 6 h were based on the satellite-derived emissions studies described above. Sensitivity tests were run for both the upper and lower range values; the largest deviation from the default setting was used to estimate the uncertainty. For $NH_3$, the uncertainty due to the lifetime is 47.9%, based on the 1.5 h lower range life-time; for $NO_2$, the uncertainty is 34.9%, based to the lower range value of 1 h. This is larger than the sensitivity calculated by Mebust et al. (2011) because, although the emissions estimate methodology is similar, a larger range of possible lifetimes is used in the present study. Other emissions estimate methods can yield larger sensitivity to lifetime; for example, Tanimoto et al. (2015) found that $NO_x$ emissions estimates vary by a factor of 2-3 depending on whether a 2 h or 6 h lifetime was assumed when working from monthly average gridded satellite $NO_2$ measurements.

## 4.4    Effect of Smoke on VCDs

At infrared wavelengths, smoke aerosol does not have a strong effect on retrievals. Clarisse et al. ( 2010a) showed the impact of biomass burning aerosol on IASI retrievals in the 800-1200 cm$^{-1}$ range and found the strongest effect for 1000-1200 cm$^{-1}$. IASI $NH_3$ is retrieved in the 800-1200 cm$^{-1}$ range, but most of the weight is for 900-980 cm$^{-1}$. CrIS retrievals are performed for 650-1095 cm$^{-1}$, with the main $NH_3$ absorbing spectral region at 960-970 cm$^{-1}$. The effect of biomass burning for 900-980 cm$^{-1}$ is small (Clarisse et al., 2010a) and therefore will have minor influence on the IASI and CrIS $NH_3$ retrievals. IASI CO retrievals are performed in the 2128-2206 cm$^{-1}$ wavelength range, where the effect of aerosol is also weaker than in the 1000-1200 cm$^{-1}$ range (Sutherland and Khanna, 1991). Therefore, uncertainties due to the effect of smoke on the VCDs are insignificant compared to other sources of error and were not included in the uncertainty calculations for IASI and CrIS.

OMI measurements are taken at UV-visible wavelengths and therefore the scattering of sunlight to the instrument is affected by the smoke plume. The ECCC AMFs (McLinden et al., 2014) are calculated assuming no aerosol is present. To account for the smoke, AMFs were recalculated using the same inputs but with an aerosol profile present as defined by a profile shape, a total aerosol optical depth (AOD), refractive index, and size distribution. The refractive index (1.5+0.01i) (Dubovik et al., 2000) and size distribution (log-normal; r = 0.1 μm, σ = 0.3) were fixed. A default profile shape with constant number density (or extinction) between 0-3 km was used, although other profile shapes were also tested. AMFs were then calculated for AODs of 0, 0.03, 0.1, 0.2, 0.5, and 1. Figure 6 shows sample AMF profiles for various AODs assuming a 0-3 km smoke plume. For

larger AODs, sensitivity to the surface is somewhat reduced and sensitivity to layers of the atmosphere above 1-2 km is somewhat higher.

The original (ECCC) $NO_2$ VCD was then used as proxy for smoke AOD using a relationship derived during the BOReal forest fires on Tropospheric oxidants over the Atlantic using Aircraft and Satellites (BORTAS) campaign (Bousserez et al., 2009):

$$AOD = (VCD - VCD_0) \times \alpha \qquad , \qquad (6)$$

where $VCD_0 = 1\times10^{15}$ molec/cm$^2$ and $\alpha = 3\times10^{-17}$ cm$^2$/molec. Interpolating between the various pre-calculated AOD-AMF tables, the AMF corresponding to the AOD was found and used to recalculate the VCD.

The OMI $NO_2$ emission estimates using the smoke AMFs were compared against emissions estimates using the typical aerosol-free CRF-based AMFs and yielded emissions estimates that were 15.1% smaller than the CRF-based AMFs. Alternative smoke AMF profiles were also tested using the triangle plume profile (Figure 4), but the change in the plume profile did not have as significant of an effect on the results (< 5% difference in emissions estimate sensitivity tests). Therefore, uncertainty in AMFs was set at 15.1% in Table 1. The relatively modest impact of the smoke on the AMF is a result of offsetting effects. There was a significant reduction in AMF due to the combination of aerosol and weighting of the $NO_2$ profile towards the surface, but this was offset by using a CRF of zero whenever the original VCD exceeded $1\times10^{15}$ molec/cm$^2$. In all cases, the shape of the aerosol profile matched that of the $NO_2$ profile.

## 4.5    Conversion of $NO_2$ to $NO_x$

In order to convert the $NO_2$ emissions estimates from OMI to $NO_x$, a factor of $NO_2/NO_x$ of 0.8 was used. This was based on an average of over 40 wildfire plumes in Alberta and Saskatchewan in 2017 using the FireWork emissions simulation with the Global Environmental Multi-scale – Modelling air quality and Chemistry (GEM-MACH) chemical transport model. Within 100 km downwind of the fire, the modelled ratio was found to be 0.8, and is consistent with measurements of $NO_2/NO_x$ at the Fort McMurray surface air monitoring stations. Other studies of $NO_x$ emissions from plumes have used values of 0.75 (Schreier et al., 2014; Tanimoto et al., 2015) and 0.7 (Mebust and Cohen, 2014; Mebust et al., 2011). Therefore, the uncertainty due to conversion from $NO_2$ to $NO_x$ in Table 1 was set to 10%. Note that $NO_x$ emissions are presented as mass of NO throughout this paper.

## 4.6    Diurnal Variation of Emissions

Emissions from wildfires vary diurnally, with more active burning and larger emissions typically occurring during the daytime. The satellite instruments measure emissions at a fixed time of day, depending on the satellite's orbit, which is assumed in this study to represent the daily average emissions rate. In order to estimate uncertainty due to diurnal variation of emissions, the diurnal weight profile from the Western Regional Air Partnership (Western Regional Air Partnership, 2005), which is used in the FireWork model, was considered. Figure 7 shows the hourly weight divided by the 24-hour weight plotted as a function of LST, with the satellite overpass time indicated. Note that the satellite overpass time does not coincide exactly with the time of the measured emissions; VCDs measured both over the hot spots (approximately at the time of emission) as well as downwind (typically ~1-2 after emission) are used together to estimate emissions. Emissions are at a minimum overnight and therefore the CrIS night-time overpass samples approximately 0.7 times the 24-hour average. Over the course of the morning, emission rates increase. Measurements taken during the IASI morning overpass are approximately equivalent to the 24-hour average. By afternoon, emissions estimates at the CrIS and OMI overpass times are 1.2× the 24-hour average. At the IASI evening overpass time, emissions estimates are similarly 1.2× the 24-hour average. Based on this diurnal profile, an uncertainty of 20% was assigned for the diurnal variation of emissions.

## 4.7    VCD from Satellite Retrievals

Uncertainties in the satellite VCD retrievals were also included in the total uncertainty. The emissions estimates are directly proportional to the mass which is calculated from the VCDs (Eq. 5) and therefore the percent uncertainty in the satellite VCD was applied directly as an uncertainty term. Note that this is based on the assumption that the satellite VCDs include systematic uncertainties that do not cancel out as data are averaged. For IASI CO, a VCD uncertainty of 35% was estimated, based on comparisons between IASI and ground-based spectrometers under fire conditions (see Sect. 2.1). . IASI $NH_3$ are ~40% lower than ground-based FTIR VCDs (Dammers et al., 2017). CrIS $NH_3$ VCDs have very little bias compared with FTIR (~0-5%) (Dammers et al., 2017) and therefore the VCD total error in the in the retrievals was used; for enhanced VCDs (greater than $2x10^{16}$ molec/cm$^2$) within a box around the fires (55-58°N and 108-114°W), the mean VCD total error was 16.5%. For OMI $NO_2$, estimated uncertainties are 30% over polluted areas with clear-sky conditions (Bucsela et al., 2013). Uncertainties in OMI VCDs due to the smoke plume have been calculated separately in Sect. 4.4.

## 5    Satellite Emissions Estimates and Comparison with Models

### 5.1    Timeseries of emissions estimates and comparison with models

Figure 8 shows the timeseries of FRP and emissions estimates for CO, $NH_3$, and $NO_x$ from satellite and from the GFAS and FireWork models during the Horse River fire. The FRP is shown for both the MODIS-Terra (morning overpass), MODIS-Aqua (afternoon overpass) and the GFAS model, which assimilates MODIS data. The FRP values from the MODIS overpasses have a similar day-to-day variability compared with GFAS, but have much larger peaks than the model. All three FRP datasets show three periods of active burning for approximately 5-8 May, 15-18 May, and 23-26 May. For CO and $NH_3$, the satellite data agree within uncertainties with the modelled data. During the first active period (5-8 May), satellite emissions agree better with the larger emissions in GFAS. During the third active period (23-26 May), the satellite emissions are closer to the lower emissions from FireWork. During the second active period (15-18 May), satellite emissions are close to both the GFAS and FireWork models. For $NO_x$, GFAS emissions are much larger than for OMI and FireWork. 24-hour average surface concentrations of CO, $NH_3$ and $NO_x$ in Fort McMurray are also enhanced during the active fire periods. The surface air monitoring data tracks somewhat with the satellite and model emissions data, with similar periods of active burning. The air monitoring station observes a larger relative peak around May 4, likely because the fire was active in Fort McMurray near the air monitoring stations at this time (MNP LLP, 2017). The smaller relative peaks in the surface air monitoring data for the third active period (23-26 May) is likely because the fire had moved to the east of Fort McMurray (MNP LLP, 2017) and therefore was further from the monitoring stations.

Table 2 shows statistics for the satellite emissions compared with the daily model output. Note that all available IASI emissions estimates (IASI-A morning, IASI-B morning, IASI-A night, IASI-B night) were averaged daily, yielding single sets of IASI emissions estimates for CO and $NH_3$. The CrIS comparisons exclude the single night-time emissions estimate. For CO, the satellite estimates fall between the GFAS and FireWork estimates, with GFAS biased high and FireWork biased low, both by ~40 kt/d. For $NH_3$, emissions estimates for GFAS and the satellite data agree well in magnitude, while the FireWork emissions estimates are ~1.5 kt/d lower than the satellite data. For $NO_x$, GFAS estimates ~5 kt/d more $NO_x$ than OMI, while FireWork emissions are much closer to the OMI measurements (within 0.5 kt/d). For all three species (CO, $NH_3$, and $NO_x$), the Pearson correlation coefficients are between 0.41 and 0.69 between satellite data and both models.

### 5.2    Emission ratios: relationships between CO, $NH_3$, and $NO_x$

Figure 9 shows the relationships between CO, $NH_3$, and $NO_x$ in the satellite and model emissions estimates and surface air monitoring station data. The surface air monitoring station and MAML measurements are shown for May 2016 only and are presented as a mass ($\mu g/m^3$) for consistency in ratios with emissions, which are calculated as a mass rate. Background levels were subtracted from the surface air monitoring concentrations. Background values were the mean of May measurements at the Fort McMurray and Fort McKay stations for 2013-2015, with hours influenced by wildfire smoke removed (Ross et al.,

2018), and were 92 ppb for CO, 0.5 ppb for $NH_3$, and 1 ppb for $NO_x$. Note that $NO_x$ measurements with more than 10 ppb of NO were excluded from the scatter plots because they often occurred for very low CO concentrations and skewed the scatter plots. The high levels of NO indicate close proximity to a $NO_x$ source, perhaps due to nearby flaming of the wildfire or anthropogenic sources in Fort McMurray. The slopes for emission and ambient concentration ratios were calculated using a linear least squares fit that was constrained to go through the origin. The 95% confidence intervals in the slope were calculated by bootstrapping the data using the bias corrected and accelerated percentile method (MathWorks, 2018).

Satellite emission ratios of $NH_3$/CO of 0.023 for IASI and 0.024 for CrIS are approximately 1.5 times larger than the modelled ratios. The MAML enhancement ratio of 0.042 is 2.5 times larger than the modelled ratios. Neither of the Fort McMurray air monitoring stations have collocated CO and $NH_3$ measurements. Paulot et al. (2017) measured $NH_3$/CO emission ratios with IASI of 0.016-0.027 over Alaska, but noted that their values are biased low compared with the emission factors calculated in situ by 35% because IASI measured older air masses. When scaled up by 35%, these values are 0.024-0.040, which fits within the range of satellite and surface ratios reported here. R'Honi et al. (2013) measured $NH_3$/CO from IASI for wildfires over Russia and found that the ratio varied in time from 0.01 to 0.052, which spans the range of values calculated in the present study. With two ground-based FTIRs, Lutsch et al. (2016) measured $NH_3$/CO emission ratios of 0.0173±0.0014 and 0.0189±0.0018 at two locations for an assumed $NH_3$ lifetime of 48 h over the long distances of the transport of the plume from fires in the Northwest Territories of Canada. These ratios are slightly lower than the values calculated in the present study; when Lutsch et al. (2016) assume a shorter lifetime of 36 h, emission ratio estimates at the two stations increase to 0.0471±0.0039 and 0.0311±0.0029, and therefore are in broad agreement with the values reported here.

For $NO_x$/CO, the FireWork and GFAS $NO_x$/CO emission ratio are 1.5 and 3 times larger, respectively, than the satellite emission ratio of 0.01. Similarly, for $NO_x$/$NH_3$ mean ratios from FireWork and GFAS are 3 and 6 times larger, respectively, than the satellite emissions ratios of 0.3-0.4. Enhancement ratios from surface stations are 0.003-0.004 for $NO_x$/CO and of ~0.06-0.08 $NO_x$/$NH_3$, which is much lower than the satellite and model values. This may be because of the weaker correlation between $NO_x$, which is released primarily during flaming combustion, and $NH_3$ and CO, which are released primarily during smouldering combustion. Furthermore, the life-time of $NO_x$ is shorter than for $NH_3$ or CO, and therefore $NO_x$ may have been lost prior to detection by the surface monitors. Simpson et al. (2011) measured $NO_x$/CO enhancement ratios for boreal forest fires during BORTAS and found values of 0.0024±0.0001 for $NO_2$ and 0.0056±0.003 for NO, which falls between the surface monitoring values in the present study.

## 5.3 Satellite-derived emission factors

Emission factors can be estimated from satellite data by assuming a linear relationship between emissions estimates and FRP and then applying a conversion factor to account for the relationship between FRP and fuel consumed (e.g., Mebust et al., 2011). The relationship between satellite emissions estimates and MODIS FRP is shown in Figure 10 for all available satellite

emissions estimates, except for the single CrIS night-time emissions estimate. May 16 and May 24 were excluded from the scatter plots and fits, because the large peaks in the MODIS FRP on these dates do not correlate with similarly high values in the emissions estimates (see Figure 8), which causes reduced correlation (R = 0.2-0.6). Correlation coefficients between the MODIS FRP and the satellite datasets range from R = 0.59-0.86 when these dates are removed. The MODIS-Aqua satellite

(LST = 13:30) has the closest overpass time to CrIS (LST = 12:30), but the MODIS-Aqua FRP did not correlate well with CrIS $NH_3$ emissions (R = 0.02). Therefore, MODIS-Terra was used instead (R = 0.59). This may be because the VCDs measured by CrIS are from emissions ~1-2 hours before the overpass time (see Sect. 4.6) and therefore are closer to the MODIS-Terra overpass time (LST = 10:30). The slope of the emissions versus MODIS FRP yields the emission coefficient (g/MJ). The slope was calculated using a linear least squares fit that was constrained to go through the origin. The 95%

confidence intervals in the slopes were calculated by bootstrapping the data using the bias corrected and accelerated percentile method in MATLAB (MathWorks, 2018).

The IASI CO emission coefficient was 49 g/MJ for IASI measurements. The $NH_3$ emission coefficient was 1.2 g/MJ for IASI and 1.1 g/MJ for CrIS. The OMI $NO_x$ emission coefficient was 0.4 g/MJ, with 95% confidence intervals of 0.2-0.6 g/MJ.

Using satellite data over multiple fires, several other studies have calculated emission coefficients for $NO_x$ over large areas. Mebust and Cohen (2014) found a value of ~0.3 g/MJ for the North American boreal forest, using a similar method with OMI satellite data. Schreier et al. (2015) found a similar value for the North American boreal forest of 0.25±0.03 g/MJ using GOME-2 satellite monthly averages on a 1°×1° grid. These two values are within the 95% confidence interval in this study. Tanimoto et al. (2015) found higher values of ~1.7-5.2 g/MJ, depending on the assumed lifetime of $NO_2$, for the boreal forest

in Alaska using monthly means of GOME and SCIAMACHY satellite on a 0.25°×0.25° grid.

A conversion factor can be applied to calculate emission factors (g/kg) from emissions coefficients (g/MJ). Some satellite $NO_x$ emissions studies (e.g., Mebust and Cohen, 2014; Schreier et al., 2015) use a single conversion factor of 0.41 kg/MJ proposed by Vermote et al. (2009) for all land types. The GFAS model, however, uses conversion factors which depend on

the land cover type based on linear regressions between GFAS FRP and dry matter combustion rate from the GFED (Kaiser et al., 2012). The Horse River fire was primarily over the extratropical forest with organic soil land cover type, with a GFAS conversion factor value of 1.55 kg/MJ, which is ~4 times larger than the 0.41 kg/MJ conversion factor. Since the conversion factors are applied directly to the emission coefficients, the choice of conversion factor can lead to emission factors that differ by a factor of ~4. Therefore, two sets of emission factors were calculated for this study, using the two different conversion

factors.

Table 3 shows the emission factors calculated by satellite for both the 0.41 kg/MJ and 1.55 kg/MJ conversion factors and used in the GFAS and FireWork model datasets. The FireWork model uses separate emission factors for flaming, smouldering, and residual smouldering. The ratio of flaming / smouldering + residual was 0.2 for the Horse River fire, as calculated using fuel

consumption types from CWFIS. Effective FireWork emission factors for the Horse River fire were calculated using this ratio and the emission factors for flaming and for smouldering and residual burning. The emission factors for the boreal forest from Akagi et al. (2011), which are based on a compilation of studies and are used in the GFED model (Van Der Werf et al., 2017) and from other studies using ground-based FTIR, aircraft, and satellite are also included for comparison.

For CO, the satellite-derived emission factors are 32 g/kg for the 1.55 kg/MJ conversion factor and 120 g/kg for the 0.41 kg/MJ conversion factor. Emission factors from GFAS (106 g/kg), the Akagi et al. (2011) compilation (127±45 kg), and the BORTAS aircraft measurements (Simpson et al., 2011) (113±17 g/kg), all fall within this range and agree slightly better with the satellite-derived emission factor for the 0.41 kg/MJ conversion factor. The emission factor used in FireWork is a somewhat larger 187 g/kg. To our knowledge, no previous studies have derived wildfire emission factors for CO using satellite.

For $NH_3$, emission factors of 0.8 g/kg and 0.7 g/kg were derived for IASI and CrIS, respectively, using the 1.55 kg/MJ conversion factor. Using the 0.41 kg/MJ conversion factor, values of 2.9 g/kg for IASI and 2.7 g/kg for CrIS were calculated. Emission factors from other studies included in the table fall within the broad range of the satellite-derived values. The emission factors used in GFAS (1.6 g/kg) and derived from ground-based FTIR (Lutsch et al., 2016) (1.40±0.72 g/kg) are between the emission factors estimated for the two different conversion factors. The emission factors used in FireWork (3.0 g/kg) and included in the Agaki et al. (2011) compilation (2.72±2.32 g/kg) are within the confidence intervals for the satellite-derived values for the 0.41 kg/MJ conversion factor.

For $NO_x$, the satellite-derived emission factors were 0.3 g/kg for the 1.55 kg/MJ conversion factor and 1.0 g/kg for the 0.41 kg/MJ conversion factor. Emission factors from the FireWork model (1.2 g/kg), the Akagi et al. (2011) compilation (0.9±069 g/kg), aircraft measurements in Alberta, Canada (Alvarado et al., 2010) (0.90±0.69 g/kg), and the BORTAS aircraft campaign (Simpson et al., 2011) (0.97±0.12 g/kg) all fall within this range. Previous satellite-derived values from Mebust et al. (2014) of 0.609±0.079 for Boreal forests and Schreier et al. (2015) of 0.61±0.07 for North American boreal forests also fall within the range derived here. The GFAS $NO_x$ emission factor of 3.4 g/kg is higher than the satellite $NO_x$ emission factors, which is consistent with the large $NO_x$ emissions predicted by GFAS compared to satellite and FireWork (see, e.g., Table 2). The GFAS $NO_x$ emission factor is for extratropical forest, which includes both temperate and boreal forests. The inventory of Akagi et al. (2011) gives an emission factor for temperate forests of 2.51±1.02, which is much larger than the value of 0.90±0.69 for boreal forests. The GFAS emission factors are, however, within the range of values derived from satellite by Tanimoto et al. (2015) for the boreal forest in Alaska.

## 5.4    Total emissions from the Horse River wildfire

The total emissions from the wildfire were calculated from the sum of all emissions estimates over all days in May 2016 for the satellite measurements and models, as shown in the blue bars in Figure 11. Total IASI emissions were calculated using

the daily average IASI emissions, as described above. The total CrIS $NH_3$ emissions estimate includes only the daytime CrIS measurements. The satellite total emissions are a sum of days with available data, and therefore are likely underestimated. The 24-hour emissions estimates from the GFAS and FireWork models were used to estimate how gaps in sampling affect the total emissions (sum of all emissions from the Horse River fire in May 2016 divided by sum of emissions on days with valid

satellite estimates). The larger of the two modelled sampling scale factors was applied to the satellite emissions as a possible upper range value, which is indicated by the errorbars on the bar chart. For comparison, the total annual anthropogenic emissions for Alberta, including both mobile and point sources in 2016, are also shown from the Air Pollutant Emissions Inventory (APEI) (ECCC, 2018) with red bars.

For CO, satellite estimates of total emissions are $1.5 \times 10^3$ kt, which falls between the GFAS total emissions estimate of $2.6 \times 10^3$ kt and the FireWork total emissions of $0.9 \times 10^3$ kt. The emissions from this fire are comparable to the total annual CO emissions from all anthropogenic sources in Alberta of 992.6 kt. Gaps in the data set have a small effect on the satellite-derived total emissions, with the upper range value of $1.7 \times 10^3$ kt based on the modelled sampling scale factors.

Satellite estimates of total $NH_3$ emissions are 30 kt for IASI and 29 kt from CrIS, which after it is scaled using model data for gaps (errorbar) agrees with the total emissions estimated from GFAS of 40 kt. Total emissions of $NH_3$ from FireWork of 16 kt is lower than the satellite and GFAS estimates. The satellite-measured $NH_3$ emissions from the Horse River fire for May 2016 are approximately 1/5 the magnitude of total annual anthropogenic emissions in Alberta in 2016 (134.8 kt).

For $NO_x$, satellite-derived total emissions estimates of 7 kt are on the same order of magnitude as FireWork total emissions of 16 kt, but are much smaller than the total emissions from GFAS of 84 kt. However, there are fewer valid days with satellite emissions estimates for $NO_x$, and therefore the modelled sampling bias is larger and more uncertain, with GFAS suggesting that 61% of the total emissions were sampled by satellite, and FireWork suggesting that only 19% of total emissions were sampled by the satellite. Therefore, the total $NO_x$ emissions as estimated from satellite could be as high as 40 kt after applying

the sampling correction. The total satellite emissions of 7-40 kt for May 2016 are about 1-7% of the magnitude of total annual anthropogenic emissions in Alberta for the year 2016. The $NO_x$ emissions from the Horse River fire are, however, on the same order of magnitude as the largest point source emitter in the Canadian oil sands area, the Syncrude Upgrader, which emitted 14 kt of $NO_x$ in 2016 (ECCC, 2017).

## 6    Conclusion

Emissions of CO, $NH_3$, and $NO_x$ from the Horse River fire were estimated using the IASI, CrIS, and OMI satellite instruments, building upon the method of Mebust et al. (2014). Our approach used satellite-derived estimates of lifetimes for $NH_3$ and $NO_x$ of 3 h and 1.5 h. Multiple datasets, including satellite (CrIS $NH_3$, CALIPSO, and MISR) and surface measurements, were

used to estimate the plume profile shape, which is used to weight the wind profiles for the emissions calculations. Based on sensitivity tests, uncertainties in the daily emissions estimates were 67% for IASI CO, 82% for IASI $NH_3$, 74% for CrIS $NH_3$, and 73% for OMI $NO_x$. Emission ratios for $NH_3/CO$, $NO_x/CO$, and $NO_x/NH_3$ were calculated from the satellite-derived and model emissions estimates and enhancement ratios were calculated from the surface concentration measurements. The ratios were compared with the literature. Broad agreement was observed for $NH_3/CO$. For $NO_x/CO$ and $NO_x/NH_3$, values ranged by a factor of 10, perhaps due to uncertainties in emission factors and the short life-time of $NO_x$.

Emission factors for CO, $NH_3$, and $NO_x$ were estimated from the satellite-derived emissions using the relationship between the daily satellite measurements and assimilated FRP from GFAS. There is large uncertainty in conversion factors, which could range from a global value of 0.41 kg/MJ (Vermote et al., 2009) or 1.55 kg/MJ (Kaiser et al., 2012), specific to the extratropical forest with organic soil land cover type. Therefore the range of satellite-derived emission factors was very wide when considering the range of possible conversion factors and confidence intervals in the fits relating FRP to emissions: ~25-146 for CO, ~0.4-3.7 for $NH_3$, and ~0.1-1.5 for $NO_x$. For the most-part emission factors for the GFAS and FireWork models, the Akagi et al. (2011) compilation, and aircraft and ground-based FTIR studies all fall within or close to these ranges; the exception is the GFAS $NO_x$ emission factor of 3.4, which is much larger than the values calculated here. Similar results were found when comparing day-to-day emissions estimates from the satellite and models, with general good agreement between model and satellite (within a factor of two), except for a clear positive bias in GFAS $NO_x$ relative to FireWork and satellite emissions. The agreement between the satellite instruments, model and emissions inventories is remarkable given the uncertainties in the satellite-derived emissions and the natural variability in emissions due to, e.g., vegetation and moisture content. In the future, more robust satellite-derived estimates of emission factors could be calculated using several approaches. Comparison between aircraft measurements and satellite results could be used to assess and improve the emissions estimate approach used here. Emission factors could be derived by applying this technique to more fires and therefore better-capturing the natural variability in emissions. Also, more advanced techniques, such as model assimilation of satellite data could be used to estimate emissions and emission factors.

**Data Availability**

- IASI CO and $NH_3$ data are available through the ESPRI Data Centre: https://cds-espri.ipsl.upmc.fr
- The CrIS-FRP-$NH_3$ science data products used in this study can be made available on request (M. W. Shephard, Environment and Climate Change Canada).
- OMI $NO_2$ data are available at: https://disc.gsfc.nasa.gov
- MODIS FRP data are available at: http://modis-fire.umd.edu/index.php. EOSDIS was used to visualize MODIS True Color and fire hotspots.
- CALIPSO data are available at: http://www.icare.univ-lille1.fr/calipso

- MISR data are available at: https://www-misr.jpl.nasa.gov/getData/accessData/
- FireWork: Environment and Climate Change Canada – Wildfire Smoke Prediction System: FireWork product website https://weather.gc.ca/firework : model input/output data available upon request
- GFAS model data were accessed using the Copernicus Atmosphere Monitoring Service (CAMS): http://apps.ecmwf.int/datasets/
- Surface air monitoring station data were collected by the Wood Buffalo Environmental Association and are available at: http://airdata.alberta.ca/
- MAML surface air monitoring data were collected by Alberta Environment Parks and are provided in the supplementary information. Data collected by the MAML on other campaigns are available upon request at: AEP.EMSD-AirshedSciences@gov.ab.ca

**Author Contributions**

CA and CM worked on the emissions estimate methodology. MS, ND, ED, JC, PM, KC-P, NT, SK, LL, and NK provided and helped to interpret the satellite, model, and ground-based datasets. The paper was prepared by CA and all authors contributed to the discussion and revision of the paper.

**Acknowledgements**

Thank you to Greg Wentworth and Matt Landis for useful conversations regarding available surface air monitoring station datasets, and to Simon Whitburn and Lieven Clarisse for information about the effect of smoke on IASI $NH_3$ and CO retrievals. Thank you to Zheng Yang for reading and commenting on the manuscript. Thank you also to the two anonymous reviewers, whose comments helped to improve the manuscript. IASI has been developed and built under the responsibility of the "Centre national d'études spatiales" (CNES, France). It is flown on-board the Metop satellites as part of the EUMETSAT Polar System." The authors acknowledge the Aeris data infrastructure (https://iasi.aeris-data.fr/nh3/) for providing access to the IASI-$NH_3$ data used in this study. Thank you to Marty Collins, Shane Taylor, and Charlene Puttick for collecting the MAML data during the wildfire.

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

**Tables**

Table 1: Summary of analysis settings and associated uncertainty for satellite emissions estimates.

| | Analysis Setting | Uncertainty IASI CO (%) | Uncertainty IASI NH$_3$ (%) | Uncertainty CrIS NH$_3$ (%) | Uncertainty OMI NO$_x$ (%) |
|---|---|---|---|---|---|
| **Method** | Flux of mass within rectangle enclosing fire | 48.8 | 48.8 | 48.8 | 48.8 |
| **Plume profile shape** | Box from 0-3 km | 22.8 | 12.7 | 12.7 | 11.2 |
| **Lifetime** | Calculated from exponential fits of downwind decay ($\tau = 3$ h for NH$_3$; $\tau = 1.5$ h for NO$_2$) | -- | 47.9 | 47.9 | 34.9 |
| **AMF** | AMFs calculated for 0-3 km plume | -- | -- | -- | 15.1 |
| **NO$_2$/NO$_x$** | Estimated using GEM-MACH model data = 0.80 | -- | -- | -- | 10.0 |
| **Diurnal variation of emissions** | No correction added | 20.0 | 20.0 | 20.0 | 20.0 |
| **VCD from satellite retrievals** | Estimates from the literature | 35.0 | 40.0 | 16.5 | 30.0 |
| **Total uncertainty** | Added in quadrature | **67** | **82** | **74** | **73** |

Table 2: Comparison statistics for emissions estimates from satellite versus models. Uncertainty in the mean difference is the standard error.

| | Mean satellite-derived emissions (kt/d) | Mean difference model minus satellite (kt/d) | | Correlation coefficient R | | Number of coincident days | |
|---|---|---|---|---|---|---|---|
| | | GFAS | FireWork | GFAS | FireWork | GFAS | FireWork |
| **IASI CO** | 88 | 42±28 | -44±13 | 0.46 | 0.57 | 17 | 16 |
| **IASI NH$_3$** | 2.3 | 0.0±0.6 | -1.4±0.3 | 0.41 | 0.69 | 14 | 13 |
| **CrIS NH$_3$** | 2.1 | 0.3±0.5 | -1.7±0.4 | 0.61 | 0.57 | 12 | 11 |
| **OMI NO$_x$** | 0.8 | 4.9±1.5 | -0.5±0.2 | 0.57 | 0.41 | 9 | 8 |

**Table 3: Emission factors from satellite, models and previous studies. For the satellite data, 95% confidence intervals from the emissions versus FRP fits are given in brackets. The FireWork emission factors are based on the mean ratio of flaming versus smouldering and residual burning for the Horse Rive fire.**

| | Method | CO EF (g/kg) | NH$_3$ EF (g/kg) | NO$_x$ EF (g/kg) |
|---|---|---|---|---|
| This study (conversion factor = 1.55 kg/MJ) | Satellite | 32 (25, 39) | 0.8 (0.6, 1.0) – IASI 0.7 (0.4, 0.9) – CrIS | 0.3 (0.1, 0.4) |
| This study (conversion factor = 0.41 kg/MJ) | Satellite | 120 (95, 146) | 2.9 (2.2, 3.7) – IASI 2.7 (1.5, 3.4) – CrIS | 1.0 (0.5, 1.5) |
| GFAS | Used in model | 106 | 1.6 | 3.4 |
| FireWork | Used in model | 187[a] | 3.0[b] | 1.2[c] |
| Akagi et al. (2011)[d] | Compilation | 127±45 | 2.72±2.32 | 0.9±0.69 |
| Simpson et al. (2011)[e] | Aircraft | 113±17 | | 0.97±0.12 |
| Lutsch et al. (2016)[f] | Ground-based FTIR | | 1.40±0.72 | |
| Alvarado et al. (2010)[g] | Aircraft | | | 0.90±0.69 |
| Schreier et al. (2015)[h] | Satellite | | | 0.61±0.07 |
| Tanimoto et al. (2015)[i] | Satellite | | | 1.11±0.23 ($\tau$=6 h) 3.34±0.69 ($\tau$=2 h) |
| Mebust et al. (2014)[j] | Satellite | | | 0.609±0.079 |

[a] From combination of CO EFs: flaming = 71.8 g/kg; smouldering and residual = 210.12 g/kg

[b] From combination of NH$_3$ EFs: flaming = 1.21 g/kg; smouldering and residual = 3.41 g/kg

[c] From combination of NO$_x$ EFs: flaming = 2.42 g/kg; smouldering and residual = 0.91 g/kg

[d] Boreal forest

[e] Plumes measured in Saskatchewan and Ontario, Canada

[f] Plumes originating from Northwest Territories, Canada

[g] Plume in Alberta, Canada

[h] North American boreal forest

[i] Alaskan boreal forest

[j] Global estimate for boreal forest

**Figures**

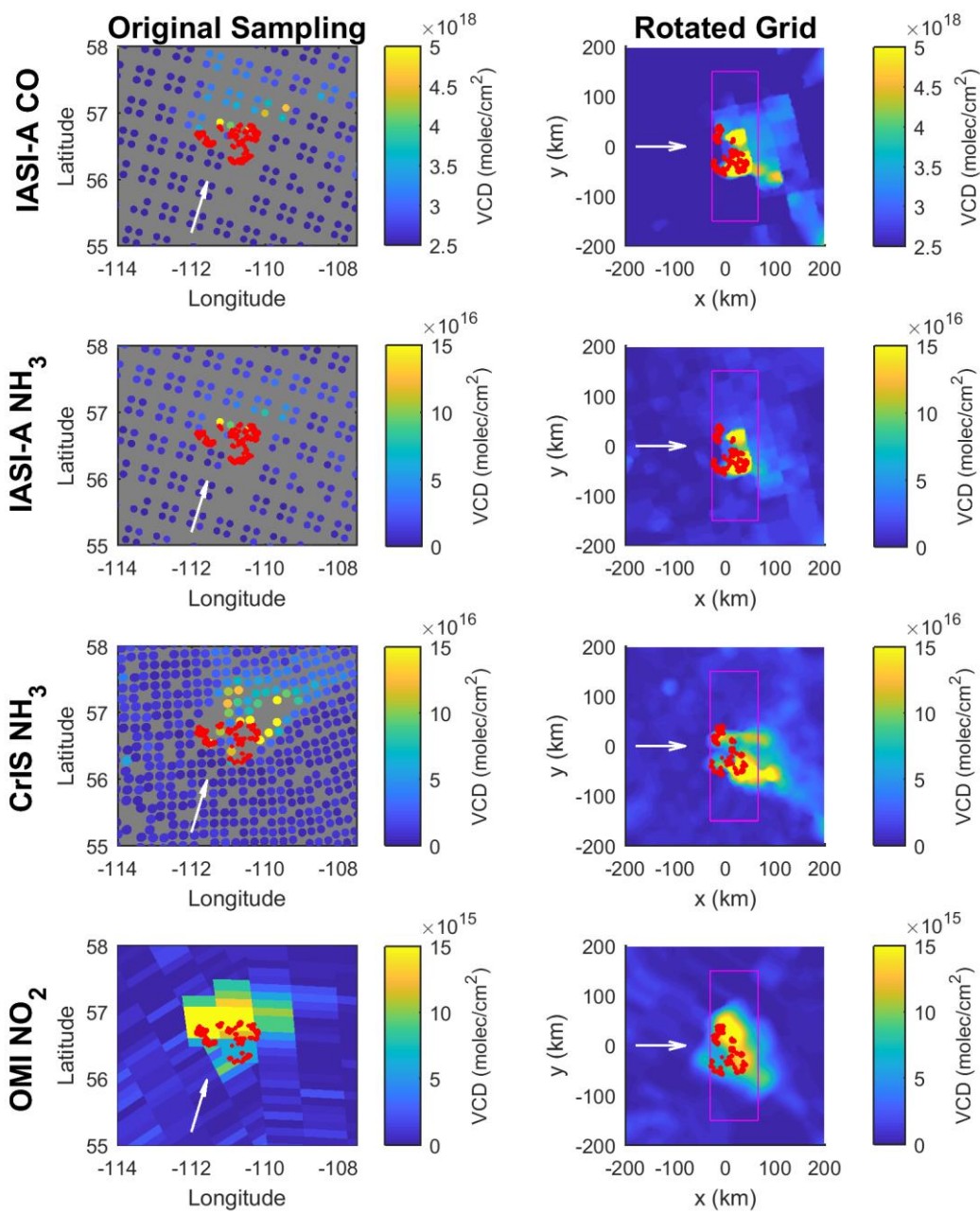

**Figure 1: Example of the analysis methods for 16 May 2016, with MODIS fire hotspots indicated (red dots). (Left) Satellite data are shown with original pixel sampling, with the ECMWF wind direction indicated by the white arrow. (Right) Data are gridded and oriented so that +x is downwind. The pink rectangles indicate the area over which the emissions are calculated.**

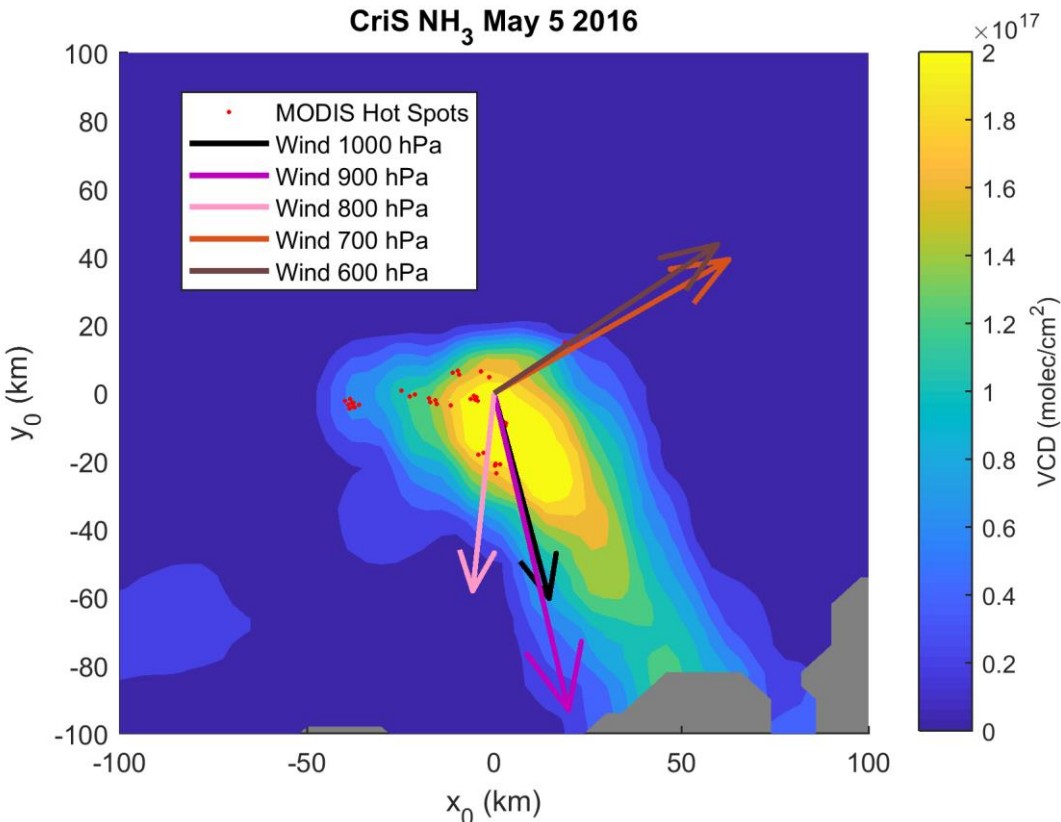

5  **Figure 2: Gridded CrIS NH₃ VCDs on 5 May 2016, with MODIS Aqua fire hot spots (red dots) and ERA5 wind directions at various altitudes (arrows). Note that VCD data are presented in an unrotated frame of reference.**

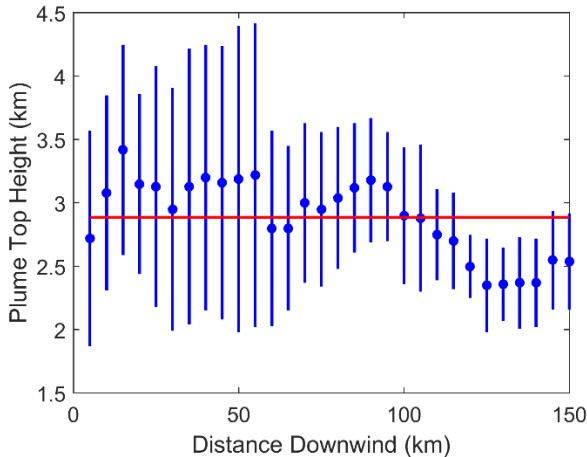

**Figure 3: MISR plume top height binned by distance downwind of the fire with 1-σ standard deviation given by the error bars. The red line indicates the mean of the binned plume top height values (2.9 km).**

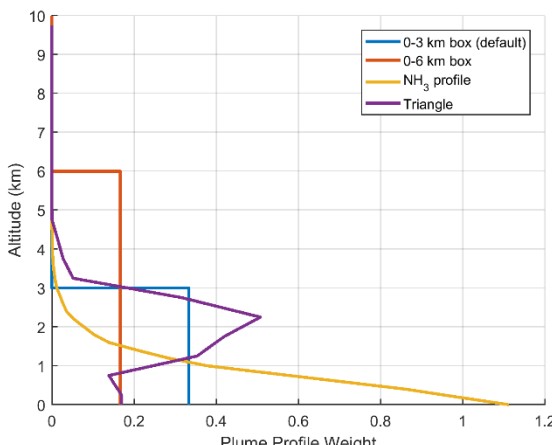

**Figure 4: Smoke plume profiles used in the emissions estimate sensitivity tests.**

5    5

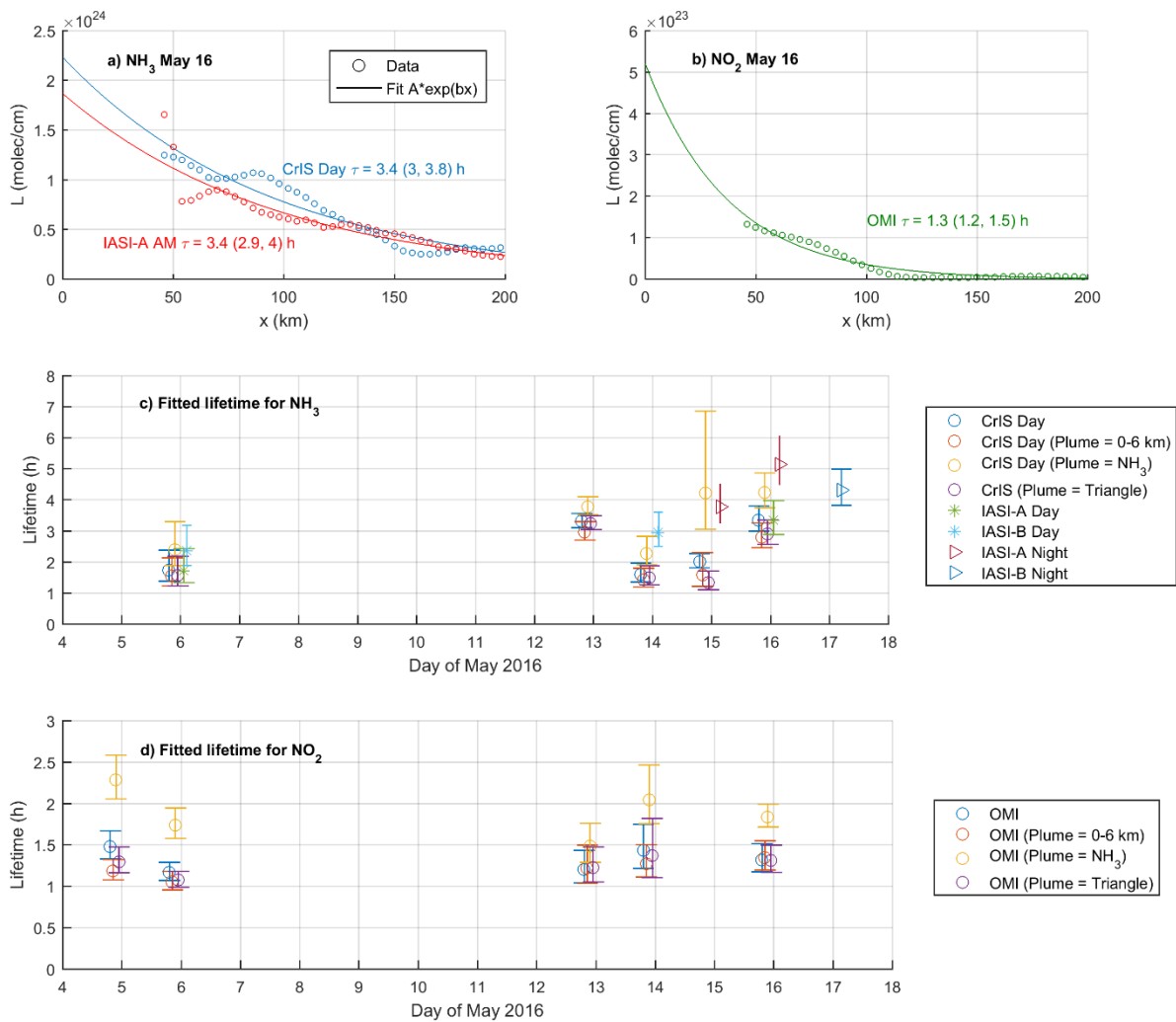

**Figure 5: Fitted lifetimes for NH₃ and NO₂. Example of fits on 16 May for (a) NH₃ from IASI morning and CrIS daytime and (b) NO₂ from OMI, with lifetime fit and 95% confidence intervals given in the text box. Lifetimes and 95% confidence intervals from fits for (c) NH₃ and (d) NO₂. Note that in panels (c), (d) markers are offset from each other in the x-axis to improve readability of the figure.**

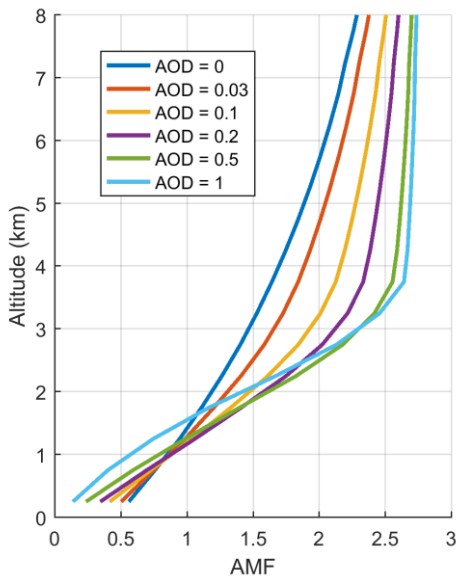

**Figure 6: Example AMFs as a function of altitude for various AODs for a 0-3 km smoke plume. AMFs are shown for clear skies above the plume, SZA = 30°, viewing zenith angle = 50°, albedo = 0.03, surface pressure = 1000 hPa, and column ozone = 350 DU.**

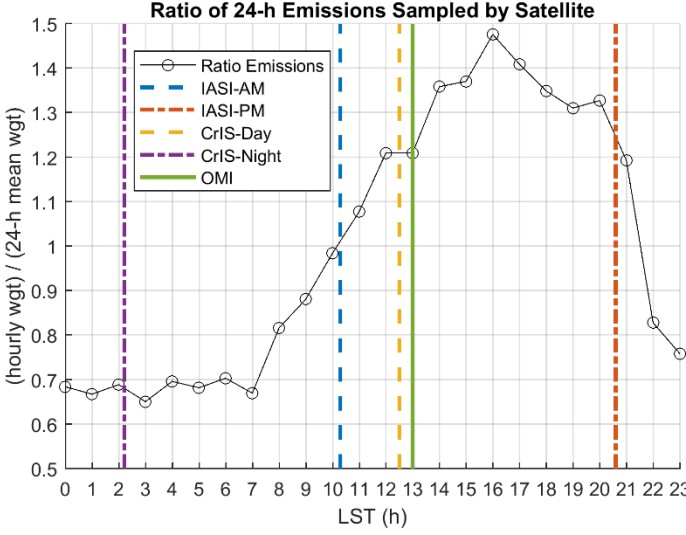

**Figure 7: Ratio of 24-hour emissions rate sampled by satellite at various LST. The y-axis shows the ratio of the hourly emissions weight over the 24-hour mean emissions weight. The typical overpass times for the IASI, CrIS, and OMI satellites over the Horse River fire area are also shown.**

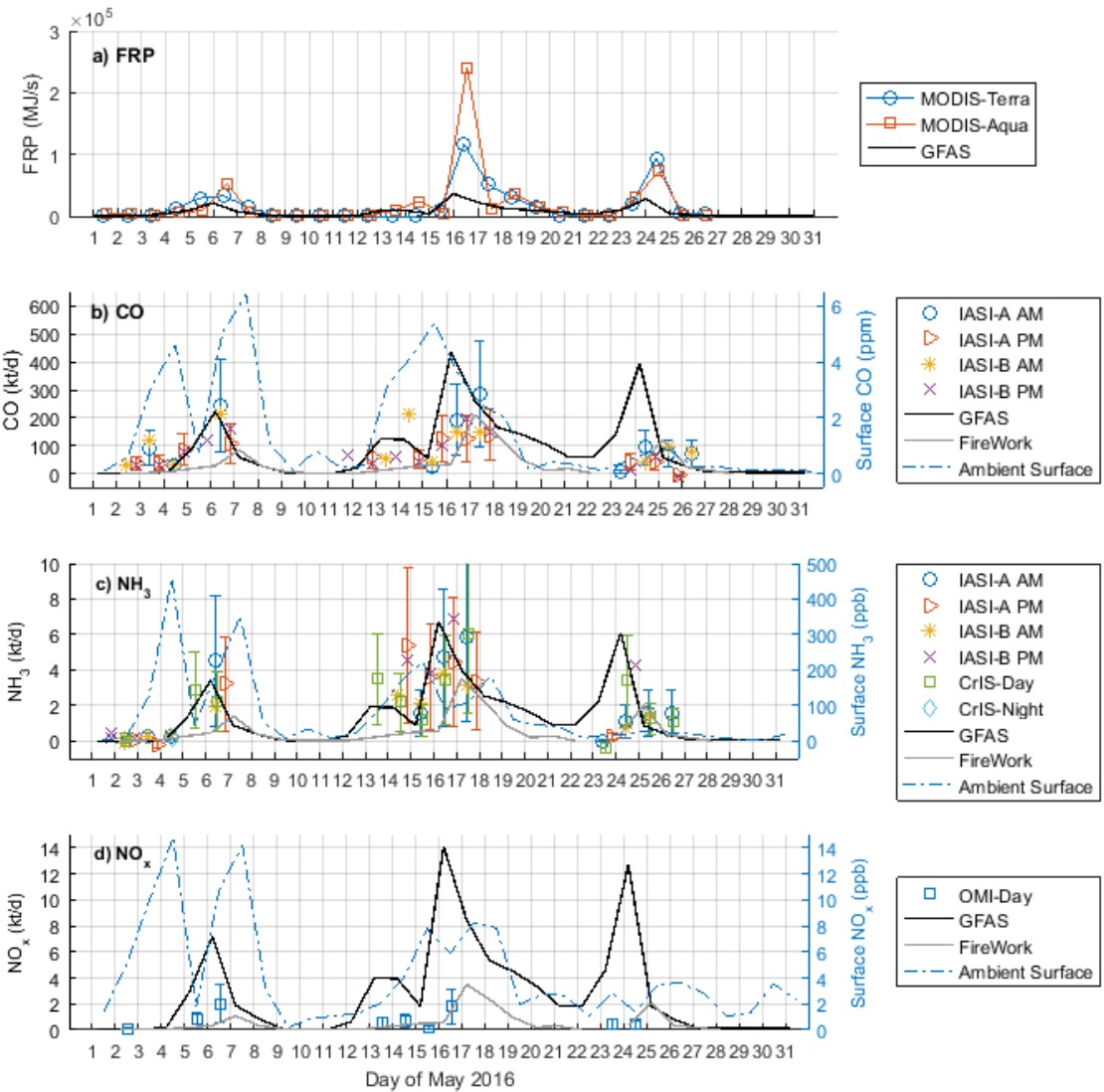

**Figure 8: Timeseries of (a) FRP and emissions estimates and surface concentrations for (b) CO, (c) NH₃, and (d) NOₓ from satellite, models and air monitoring stations. Note that errorbars are not shown for IASI-B emissions to improve readability of the figure. The surface concentrations are the 24-h average concentrations of CO at Fort McMurray – Athabasca Valley air monitoring station, and NH₃ and NOₓ at Fort McMurray – Patricia McInnes air monitoring station. Note that NOₓ measured at Fort McMurray – Athabasca Valley shows similar variability, but is not shown here.**

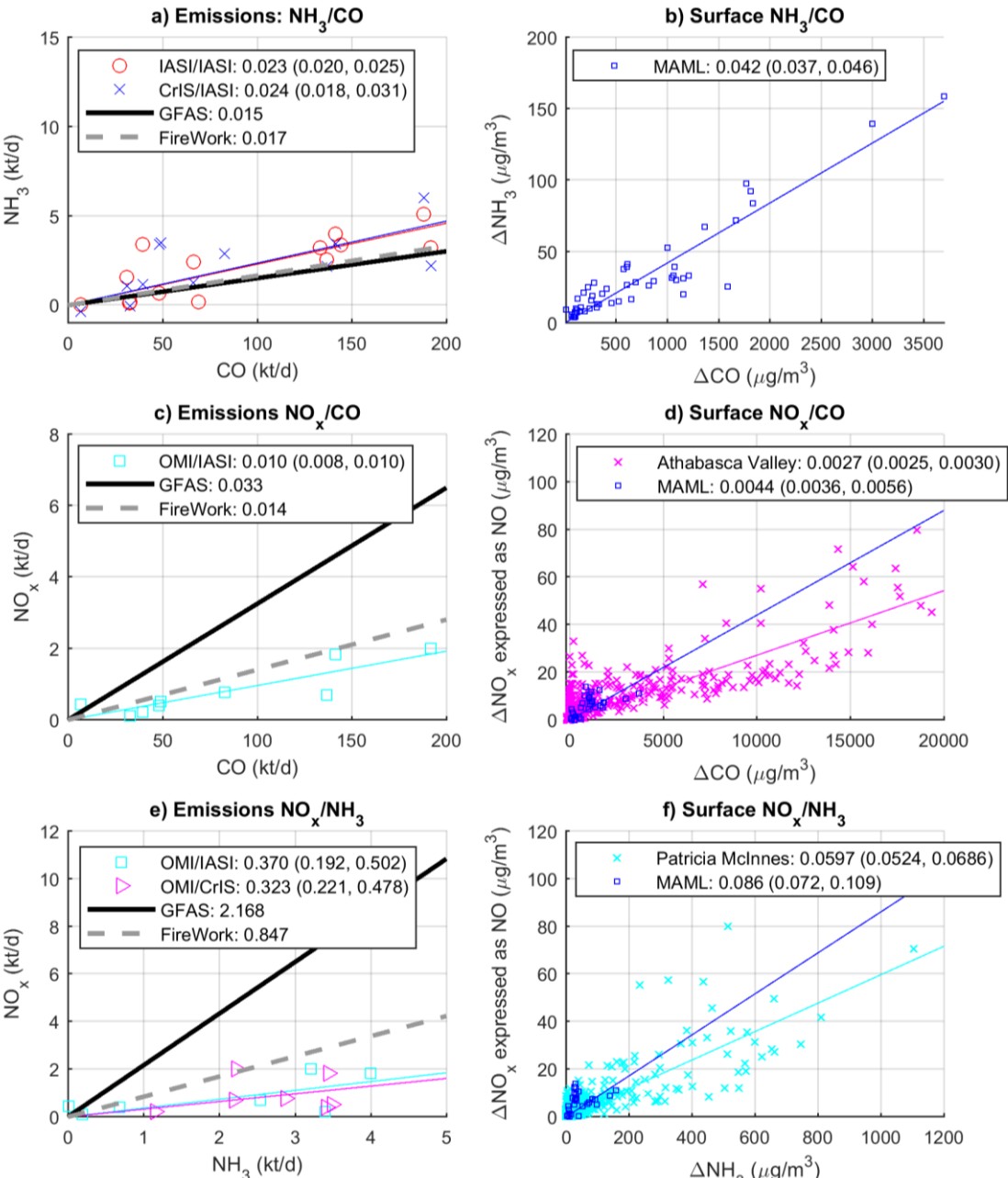

**Figure 9: Scatter plots of emissions (left) and surface concentrations (right) for NH₃/CO (top), NOₓ/CO (middle), and (bottom) NOₓ/NH₃. Daily emissions are shown for satellite and the GFAS and FireWork models. The difference in surface concentrations are hourly averages minus background concentrations. The slope and 95% confidence interval, given in brackets, of a linear fit, constrained to go through the origin, are also shown in the legend.**

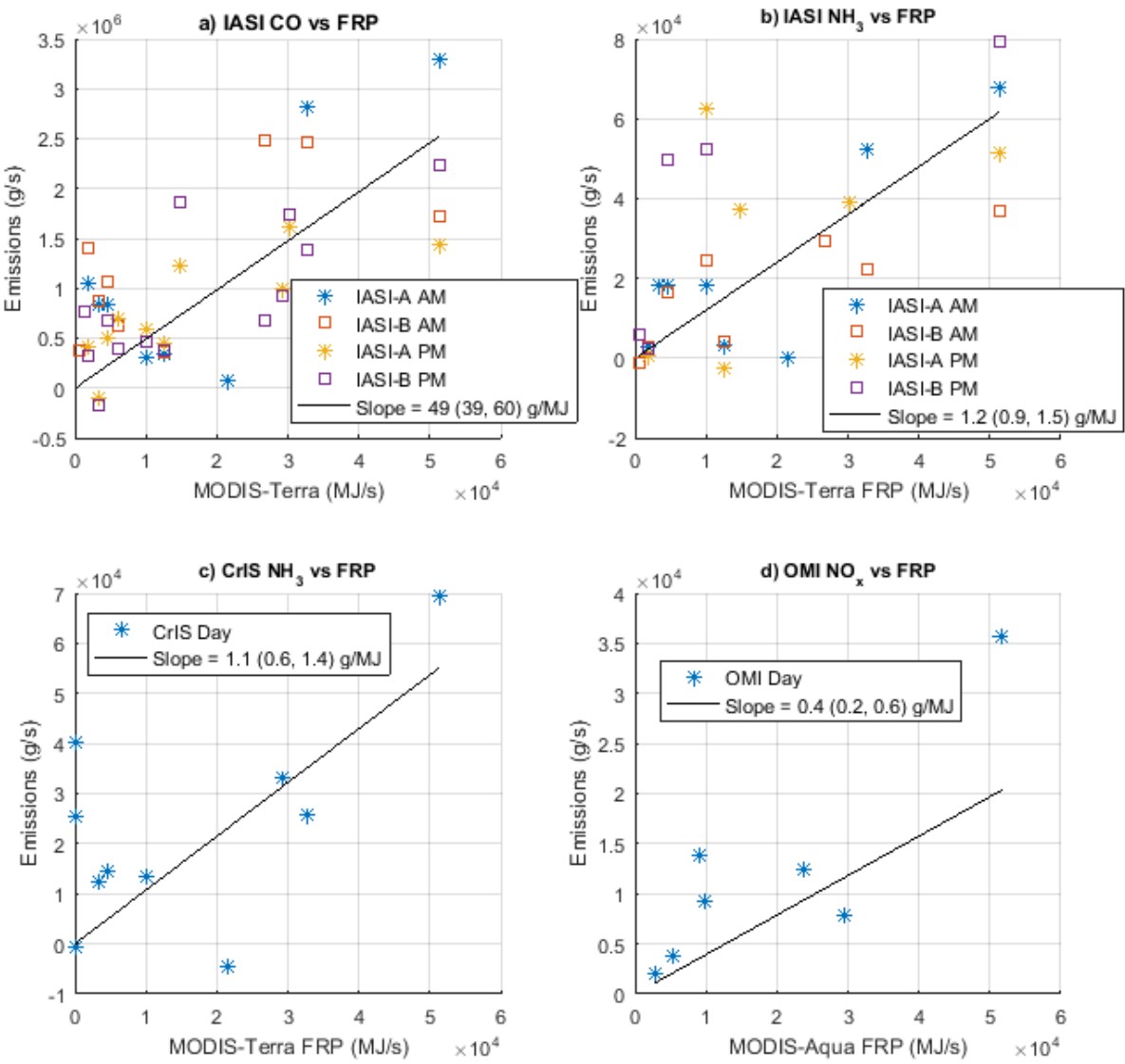

**Figure 10: Scatter plot of daily emissions versus MODIS FRP for (a) IASI CO, (b) IASI NH₃, (c) CrIS NH₃, and (d) OMI NOₓ. The slope and 95% confidence interval of a linear fit, constrained to go through the origin, are also shown.**

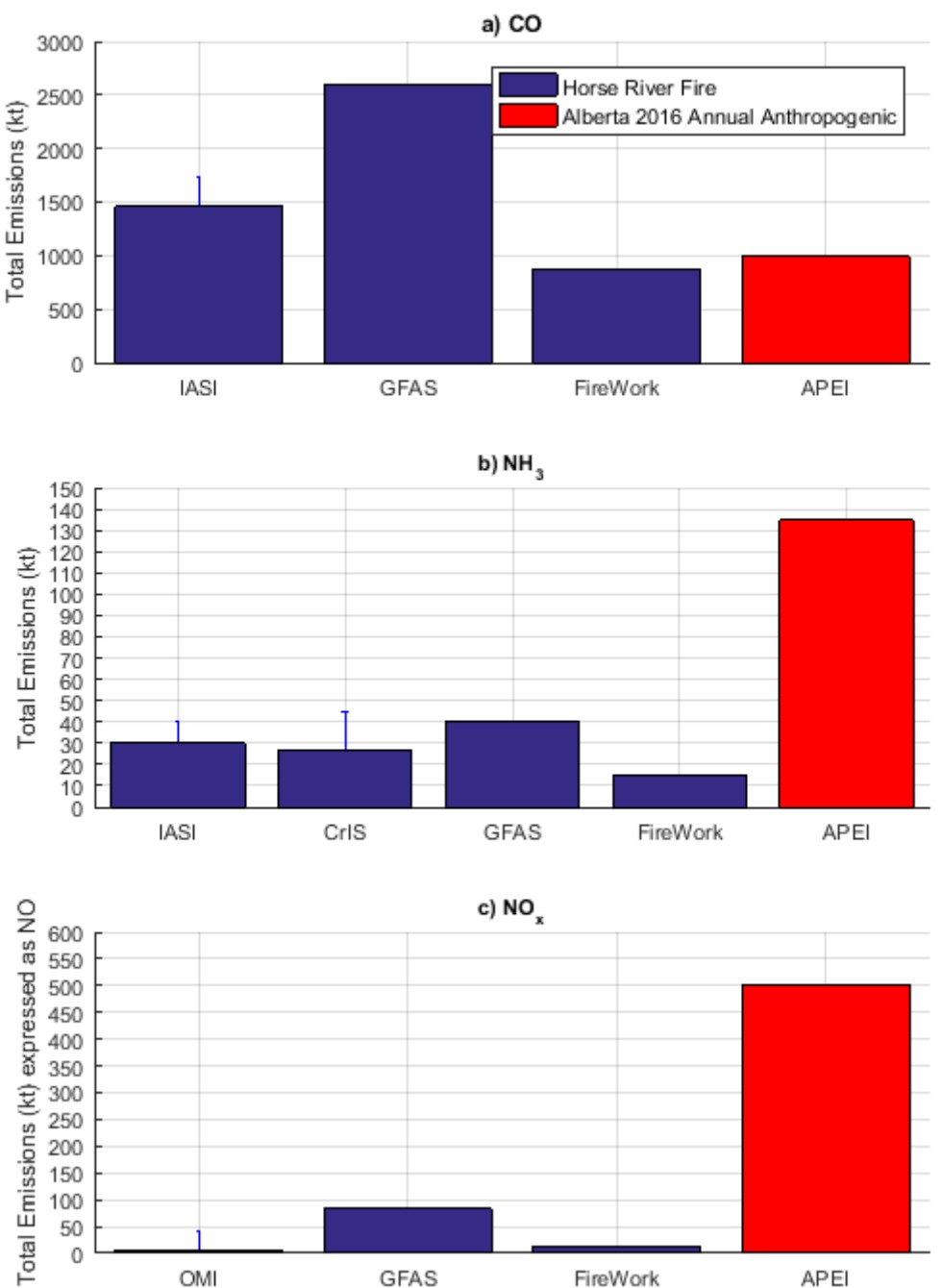

**Figure 11: Total emissions for the Horse River fire in May 2016 from satellite and models (blue), with total annual anthropogenic emissions for Alberta from the 2016 APEI (red). The errorbars represent satellite emissions scaled by models to account for gaps in sampling. The APEI emissions have been converted so that they are expressed as kt of NO.**