# Peer review of "Satellite-derived emissions of carbon monoxide, ammonia, and nitrogen dioxide from the 2016 Horse River wildfire in the Fort McMurray area"

_Atmospheric Chemistry and Physics, 2018_

## Referee Comment (RC1) · Anonymous Referee #2 · 23 Oct 2018

This is an interesting study that integrated multiple satellite observations to quantify lifetime and emissions of key species during the 2016 Horse River fire, based on fitting the Gaussian plume model. The results were interpreted in the context of an extensive suite of observations and fire models, as well as related publications. The topic is well within the scope of ACP, and the paper is in general well written and easy to follow. I found some technical details missing which I would list in the review. I recommend the manuscript to be accepted after the following concerns are addressed.

1. Page 5, Line 6-14: Please clarify the necessity and effect of increasing sampling. For example, would the number of available data be substantially reduced if the data

affected by row anomaly were not included? Would the uncertainties of NOx emission estimates in Table 1 be significantly stronger due to the reduced sampling?

2. Section 4: Apart from all the uncertainty tests, I am surprised that the authors did not do sensitivity calculations that account for the uncertainties in the satellite retrievals?

3. Section 4.1: Maybe the authors could list some uncertainties for NH3 and NOx in this section assuming the default lifetime they used later, just to provide a perspective if it could actually be larger or smaller than the 48% determined from CO?

4. Section 4.2: There is generally 1 or less than 1 piece of information in the NH3 retrieval from CrIS, and the average kernels of CrIS suggest the retrievals are most sensitive to 800-900 hpa (Shepard and Cady-Pereira, 2015). It is not surprising to me if CrIS resolves the variation of NH3 that are more reflective of winds at such altitude. Using MISR and CALIPSO plume height data is a good idea since the multi-angle capability of MISR and the LIDAR signals in CALIPSO do contain vertical information. I suggest the authors to reconsider (or further justify) the value of CrIS NH3 in determining the plume shape.

5. Page 12, Line 22-23: I am interested in the proportion of "accepted fitting", which gives a sense of the applicability of this fitting method in the data investigated. Also, for the cases with larger fitting errors, where are the errors from?

6. Page 16, Line 32-33: Instead of just omitting the MODIS data, could the authors comment on if the emission/FRP relationship might become non-linear at very strong burning conditions, and a good fit could still be achieved by deleting the two peaking records of FRP in Figure 8? The derived emission factors could still be meaningful, representing constant burning conditions.

7. Figure 10: I am interested to see if the correlation would be better or worse if the diurnal variation (e.g. in Figure 7) were corrected?

8. Since the authors made great efforts in quantifying and analyzing the uncertainties

which are insightful enough to be part of the main findings in the paper, I suggest adding a related discussion in the abstract and the conclusion. An example could be "The uncertainties of emission estimates are more sensitive to the plume shape for CO, and to the fitted lifetime for NH3 and NOx".

Technical suggestions:

1. Page 1, Line 30: "0.03" should be "0.003"?

2. Page 2, Line 21-22: Maybe change to "more than 10% of global CO emissions from wildfires are over mid- and high- latitude."

3. Page 8, Line 27-28: Please clarify how to determine if one day is with "sufficient" data, and how the "gap filling" was done?

4. Page 9, Line 15: I suppose the "VCD" here should actually be "dVCD"?

5. Page 12, Line 17: Again, how to define "sufficient"?

6. Page 16, Line 2 and Page 17, Line 3: Should add a reference to this estimation method.

7. Page 17, Line 7: Should add "for NOx" somewhere in this line to guide the presentation in the rest of this paragraph.

Shephard, M. W., and Cady-Pereira, K. E.: Cross-track Infrared Sounder (CrIS) satellite observations of tropospheric ammonia, Atmos. Meas. Tech., 8, 1323-1336, 2015.

---

## Referee Comment (RC2) · Anonymous Referee #1 · 5 Nov 2018

In this work, Adams et al. analyse wildfire events in Northern Alberta and Saskatchewan in May 2016. They utilize various satellite datasets to derive emission ratios, emission factors and total emissions of CO, NH3 and NOx. The results have been compared to several external dataset, like global fire models, emission inventories as well as in-situ surface concentrations measurements. A considerable part of the work is dedicated to the quantification of method uncertainties. As detailed below, my major concern is that errors of the satellites' total column observations have not been taken into consideration quantitatively. Nonetheless, the paper fits well into the scope of ACP and I support its publication after the comments below have been addressed appropriately.

[Figure]

Specific comments:

P4, L2: 'which showed overall good results and performance'

This sentence is very general and does not contain useful information. There should be quantitative error estimates of the total CO column amounts, which can be used for the error analysis within the present work.

P4, L6: 'and showed an overall good performance'

This part may be deleted since quantitative numbers for uncertainties of IASI NH3 total column amounts are provided subsequently.

P4, sections 2.1 and 2.2:

Please provide information about the vertical averaging kernel for each IR product. In general, mid-IR nadir sounding is not sensitive near the ground (with few exceptions in case there are large temperature differences between the surface temperature and the temperature of the atmospheric boundary layer). Thus, the information on the trace gas concentration in the lower layers, which is used to determine the total column amounts is mainly determined by the a-priori profile.

P8, section 4 and Table 1:

A chapter on the uncertainties connected to the total column observations should be added.

P9, L2-4:

For the baseline VCDs only IASI-A morning and CrIS daytime values have been used. Why not IASI-B and evening/night data? Is there any explanation, why CrIS and IASI values of NH3 baseline VCDs differ by a factor of 10?

P10, L30: 'The direction of the NH3 plume aligns best with winds between 1000-800 hPa (approximately 0-2 km), suggesting that the bulk of the NH3 plume is within this

altitude range.'

You should consider discussing the influence of the CrIS averaging kernel here, since it is not sensitive to the lowest layers.

P14, L1: 'Note that the IASI CO and NH3, and CrIS NH3 are measured at infrared wavelengths and therefore are not sensitive to the smoke plume.'

This is not entirely correct. Since smoke is absorbing at mid-IR wavelengths, there should be an influence on the results in case it is not taken into account explicitly in the satellite retrieval procedure. I would be interested if there are any sensitivity calculations for IR retrievals in presence of smoke.

Technical:

P13, L32: 'due the' -> 'due to the'

P14, L22: 'emissions' -> 'emission'

P16, L28: 'then applying conversion factor' -> 'then applying a conversion factor'

P18, L5: '1.0' -> '1.0 g/kg'

P18, L6: '3.7' -> '3.7 g/kg' and '3.9' -> '3.9 g/kg'

---

## Author Comment (AC1) · 19 Jan 2019

We thank anonymous referee #2 for their comments, which have helped to improve the manuscript. We have provided the referee comments in italics, with responses to each comment below.

*1. Page 5, Line 6-14: Please clarify the necessity and effect of increasing sampling. For example, would the number of available data be substantially reduced if the data affected by row anomaly were not included? Would the uncertainties of NOx emission estimates in Table 1 be significantly stronger due to the reduced sampling?*

In total, there were nine days with enough OMI VCDs to estimate emissions. On five of these days (May 2, 5, 14, 16, 23), the OMI measurements over and downwind of the fire hot spots were not affected by the row anomaly. On four of these days (May 6, 13, 15, 24), the row anomaly affected more than half of the measurements over and downwind of the fire hot spots.

In order to test the effect of reduced sampling, the VCDs on May 5, which were not affected by the row anomaly, were filtered to remove VCDs in a pattern similar to the row anomaly. With full sampling, emissions estimates were 1 kt/d. With the reduced sampling, emissions estimates decreased to 0.4 kt/d. This demonstrates that inclusion of the row anomaly data was important to have sufficient sampling on the four days affected by the row anomaly.

In order to reflect this, we've added the following to Sect. 2.3

"Inclusion of the row anomaly data was required to have sufficient sampling to estimate emissions for 6 May, 13 May, 15 May, and 24 May. The row anomaly did not affect VCDs over the fire hot spots for other days."

*2. Section 4: Apart from all the uncertainty tests, I am surprised that the authors did not do sensitivity calculations that account for the uncertainties in the satellite retrievals?*

We have added uncertainty terms for the satellite retrievals, as recommended. These are given in Sect. 4.7 and are included in Table 1.

*3. Section 4.1: Maybe the authors could list some uncertainties for NH3 and NOx in this section assuming the default lifetime they used later, just to provide a perspective if it could actually be larger or smaller than the 48% determined from CO?*

The difficulty with running these tests for $NH_3$ and $NO_x$ is that the alternative method tested, 20-km downwind flux, is not very good for short-lived species and yields large differences from the emissions estimate method given in the paper. This is because for typical winds at the plume altitudes ~20 km/h, VCDs 20 km downwind have aged by 1-hour, which is approaching the lifetime of $NO_x$ and is ~1/3 the lifetime of $NH_3$ for this fire. Therefore, the downwind measurements are taken at much lower levels of $NO_x$ and $NH_3$ and are sensitive to the choice in lifetime. Therefore, we applied the test for CO only and used the same value for $NH_3$ and $NO_x$. We have changed the wording in the text to better-reflect this reasoning: "The value of 48.8% method uncertainty was used for both $NH_3$ and $NO_x$ because the alternate method used in the sensitivity tests is based on downwind flux, where $NH_3$ and $NO_x$ line

densities are smaller due to their short life-times. Therefore, the alternate method is very sensitive to the assumed lifetimes and is therefore not appropriate for these species."

*4. Section 4.2: There is generally 1 or less than 1 piece of information in the NH3 retrieval from CrIS, and the average kernels of CrIS suggest the retrievals are most sensitive to 800-900 hpa (Shepard and Cady-Pereira, 2015). It is not surprising to me if CrIS resolves the variation of NH3 that are more reflective of winds at such altitude. Using MISR and CALIPSO plume height data is a good idea since the multi-angle capability of MISR and the LIDAR signals in CALIPSO do contain vertical information. I suggest the authors to reconsider (or further justify) the value of CrIS NH3 in determining the plume shape.*

The total column averaging kernels for CrIS are shown in the response to referee #1, and has values close to 1 up to ~3 km. Therefore, CrIS is sensitive to $NH_3$ at 700 hPa, and Fig. 2 therefore suggests that there is not much $NH_3$ at this altitude (~2.6 km) compared with the lower altitudes. CrIS averaging kernels show no sensitivity above 3 km, which is expected if there is little or no $NH_3$ at these altitudes.

*5. Page 12, Line 22-23: I am interested in the proportion of "accepted fitting", which gives a sense of the applicability of this fitting method in the data investigated. Also, for the cases with larger fitting errors, where are the errors from?*

The proportion of accepted fitting is given in Table 1 below. The fits were performed using data from the fire hotspots to 200 km downwind of the fires. Therefore, for a good fit, the winds must be fairly consistent over a large area. For ammonia, the primary reason for poor fits was low wind-speeds and variable wind directions downwind of the fires, which led, for example, to the accumulation of ammonia at some locations far downwind. Other poor fits occurred when there were gaps in the data far downwind, which led to poor interpolation at these locations. For $NO_2$, no fits failed the fitting error criterion. This is likely because, due to the short lifetime, there is very little $NO_2$ far downwind and therefore the fit is not as sensitive to inconsistencies in the winds.

This demonstrates that this fitting method to estimate lifetime is applicable only on days with winds that are consistent downwind of the fire. Therefore, in this paper, we did not use a daily fitted lifetime for each satellite instrument. Instead we used the fitted lifetimes with successful fits to get a best estimate of lifetime for ammonia and $NO_2$ and used other values in the literature to estimate uncertainties in the lifetime. Note that, unlike the lifetime calculations, the emissions estimates are not sensitive to variable wind conditions or data gaps far downwind because they only includes data over and 20 km downwind of the fire hotspots.

We have added the following to Sect. 4.3 to describe this:

"Approximately 60% of ammonia fits with sufficiently large plumes did not meet the fitting error criteria; this occurred primarily when wind speeds were low and/or winds were variable downwind, and therefore the plume changed direction and/or accumulated over some downwind locations. All $NO_2$ fits met the fitting criteria, as the short lifetime of $NO_2$ makes it less sensitive to inconsistencies in the winds far downwind."

*Table 1: Number of days included in lifetime estimates.*

| | # days with emissions estimates | # days with emissions estimates > thresholds (1 kt/d for $NH_3$, 0.5 kt/d for $NO_2$) | # days with emissions estimates > thresholds & fitting error < 1 h |
|---|---|---|---|
| CrIS $NH_3$ | 12 | 10 | 6 |
| IASI-A Day $NH_3$ | 10 | 7 | 3 |
| IASI-B Day $NH_3$ | 10 | 6 | 2 |
| IASI-A Night $NH_3$ | 8 | 5 | 2 |
| IASI-B Night $NH_3$ | 6 | 4 | 1 |
| OMI $NO_2$ | 9 | 6 | 6 |

*6. Page 16, Line 32-33: Instead of just omitting the MODIS data, could the authors comment on if the emission/FRP relationship might become non-linear at very strong burning conditions, and a good fit could still be achieved by deleting the two peaking records of FRP in Figure 8? The derived emission factors could still be meaningful, representing constant burning conditions.*

As suggested, we tried removing the two days with peaks in FRP (May 16 and 24) and the correlation improved significantly, so that it was better than the correlation with GFAS assimilated FRP. Therefore, we elected to replace GFAS FRP with MODIS FRP in the paper and revised Sect. 5.3, Table 3, and Fig. 10 accordingly.

*7. Figure 10: I am interested to see if the correlation would be better or worse if the diurnal variation (e.g. in Figure 7) were corrected?*

In order to test this, we scaled the emissions estimates to a 24-h mean value for the scatter plot against GFAS FRP (Fig. 7 in the discussions paper). The satellite does not measure exactly at the time of the emission – instead it measures VCDs, some of which are right over the fire hot spots (very recently emitted) and some of which are downwind. In order to account for this, the time of emission was using the average age of the air within the box (clearing time divided by 2). The corrections for diurnal variation did not improve correlation with FRP. For IASI CO, R=0.48 without corrections and R=0.42 with corrections. For IASI NH3, R=0.46 without corrections and R=0.48 with corrections. This is not surprising since the FireWork diurnal variation is based on a simple scaling of the data according to time of day and is not calculated specifically for this fire. Furthermore, many other factors affect the emissions estimates, such as the assumed life-time and the plume height, all of which could blur the effect of the diurnal variation on the relationship to FRP.

*8. Since the authors made great efforts in quantifying and analyzing the uncertainties which are insightful enough to be part of the main findings in the paper, I suggest adding a related discussion in the*

*abstract and the conclusion. An example could be "The uncertainties of emission estimates are more sensitive to the plume shape for CO, and to the fitted lifetime for NH3 and NOx".*

We have added the following to the abstract: "Sensitivity tests were performed and it was found that uncertainties of emission estimates are more sensitive to the plume shape for CO, and to the fitted lifetime for $NH_3$ and $NO_x$."

*Technical suggestions:*

*1. Page 1, Line 30: "0.03" should be "0.003"?*

This has been corrected

*2. Page 2, Line 21-22: Maybe change to "more than 10% of global CO emissions from wildfires are over mid- and high- latitude."*

This has been changed as recommended

*3. Page 8, Line 27-28: Please clarify how to determine if one day is with "sufficient" data, and how the "gap filling" was done?*

We have added the following about determining if there is sufficient data:

"This was done by visually inspecting the original and gridded VCDs for each day; if gaps in the data covered large areas that were required to resolve the plume or led to interpolation of the plume that looked suspect, the day's data was excluded.  Most of the days excluded were missing more than half of the data over or downwind of the fire hot spots."

And the following about gap filling:

"… using interpolation with the inpaint_nans function in MATLAB (D'Errico, 2009)."

*4. Page 9, Line 15: I suppose the "VCD" here should actually be "dVCD"?*

Yes, this has been changed to dVCD

*5. Page 12, Line 17: Again, how to define "sufficient"?*

This was addressed above

*6. Page 16, Line 2 and Page 17, Line 3: Should add a reference to this estimation method.*

A reference has been added

*7. Page 17, Line 7: Should add "for NOx" somewhere in this line to guide the presentation in the rest of this paragraph.*

"for NOx" has been added to clarify this

---

## Author Comment (AC2) · 19 Jan 2019

We thank anonymous referee #1 for their comments, which have helped to improve the manuscript. We have provided the referee comments in italics, with responses to each comment below.

*Specific comments:*

*P4, L2: 'which showed overall good results and performance' This sentence is very general and does not contain useful information. There should be quantitative error estimates of the total CO column amounts, which can be used for the error analysis within the present work.*

We have added quantitative information with a reference to a comparison with ground-based FTIR to this section. We have now included an uncertainty in total column in Sect. 4.7, as discussed in the comments below.

"IASI CO VCDs have also been compared against ground-based FTIR measurements, with typical differences of ~10% (Kerzenmacher et al., 2012)."

*P4, L6: 'and showed an overall good performance' This part may be deleted since quantitative numbers for uncertainties of IASI NH3 total column amounts are provided subsequently.*

This has been deleted

*P4, sections 2.1 and 2.2: Please provide information about the vertical averaging kernel for each IR product. In general, mid-IR nadir sounding is not sensitive near the ground (with few exceptions in case there are large temperature differences between the surface temperature and the temperature of the atmospheric boundary layer). Thus, the information on the trace gas concentration in the lower layers, which is used to determine the total column amounts is mainly determined by the a-priori profile.*

We agree that this information is helpful for interpreting the emissions estimates and therefore, we have added to Sections 2.1 and 2.2.

For IASI CO, sample averaging kernels for the VCD are shown in Figure 1 below. We have added the following discussion in Sect. 2.1 to address this. We accounted for this in the uncertainty estimates by applying an uncertainty of 35% to the VCDs in Sect. 4.7.

"During the fire, total column averaging kernels showed increased sensitivity at surface with values of ~0.4-0.6 CO for large VCDs greater than $3.5 \times 10^{18}$ molec/cm$^2$. These large VCD measurements are taken within the smoke plume, and are the primary contributor to the emissions estimates. This is consistent with previous studies which have found increased sensitivity to surface CO for large VCDs (Bauduin et al., 2017). Yurganov et al. (2011) compared IASI VCDs with measurements from three grating spectrometers during forest and peat fires plume in Central Russia in July-August 2010. They found that the IASI VCDs were biased low compared with the ground-based measurements by an estimate of $1.61 \times 10^{18}$ molec/cm$^2$ or ~35%, over a sample with a mean IASI CO VCD of $4.7 \times 10^{18}$ molec/cm$^2$."

[Figure]

*Figure 1: CO total column averaging kernels for IASI-A for daytime measurements on May 16, 2016 (as for Fig. 1 of the paper), with colour scale indicating the VCD.*

For IASI $NH_3$, we have added the following to the text "Averaging kernels are not produced as a part of the $NH_3$ retrievals; however, previous studies have demonstrated good agreement with surface and FTIR measurements (e.g., Clarisse et al., 2010; Van Damme et al., 2015; Dammers et al., 2017), demonstrating that there is sensitivity to the lower layers of the atmosphere."

An example of total column averaging kernels for CrIS $NH_3$ VCDs are shown in Figure 2 for May 16 2016. We have added the following to Sect. 2.2 to address this: "CrIS total column averaging kernels during the fire suggest good sensitivity to $NH_3$ in lower layers of the atmosphere, with values of ~0.5-1.5 for the 0-3

km altitude range (not shown here). Above ~3 km, the sensitivity is very low, which is expected if ammonia concentrations are low at these altitudes (see Sect. 4.2)."

[Figure]

*Figure 2: NH₃ total column averaging kernels for CrIS for daytime measurements on May 16, 2016 (as for Fig. 1 of the paper).*

*P8, section 4 and Table 1: A chapter on the uncertainties connected to the total column observations should be added.*

This has been added, as recommended (see Sect. 4.7 and Table 1).

*P9, L2-4: For the baseline VCDs only IASI-A morning and CrIS daytime values have been used. Why not IASI-B and evening/night data? Is there any explanation, why CrIS and IASI values of NH3 baseline VCDs differ by a factor of 10?*

The description of the use of baseline VCDs was not clear in the text. We calculated baseline VCDs separately for the various measurement types (e.g., IASI-A vs IASI-B, day vs night), but were describing a subset of the baseline values as an example. We have added some detail to the text in this paragraph describing the baseline VCD calculations and have expanded the text to show baseline values for all the measurement types in order to improve clarity.

As the NH$_3$ background values approach the minimum detection levels of the instruments there can differences between the IASI and CrIS retrieved products.  The detection limit depends on the instrument characteristics and atmospheric state, with a minimum detection limit of ~2-3 ppbv (~4-6x10$^{15}$ molec/cm$^2$ for IASI (Clarisse et al., 2010) and of ~0.5-1.0 ppbv (~1-2x10$^{15}$ molec/cm$^2$) for CrIS (Shephard and Cady-Pereira, 2015; Kharol et. al., 2018).  Thus CrIS will be able to provide measurement information for lower baseline amounts. However, once baseline values approach respective instrument detection limits there can be differences in the reported products depending on how the non-detects are handled.  Dammers et al. (2017) showed that IASI NN values in the lower range tend to be low biased (~50%), which might account for a lower computed baseline.  The CrIS algorithm provides the sensitivity and ability to identify non-detects, but in this study there is no attempt to account for these non-detects, and only observations with some measurement contribution (DOFS > 1.0) are included, which can result in a high bias under conditions when there is a significant fraction of retrievals below the detection limit of the instrument, for example ~-25% shown by Dammers et al., (2017), which might account for the higher baseline.

*P10, L30: 'The direction of the NH3 plume aligns best with winds between 1000-800 hPa (approximately 0-2 km), suggesting that the bulk of the NH3 plume is within this altitude range.' You should consider discussing the influence of the CrIS averaging kernel here, since it is not sensitive to the lowest layers.*

As shown in the averaging kernels above, CrIS VCDs are sensitive to the lowest layers of the atmosphere (~0-3 km) and therefore this approach is reasonable.

*P14, L1: 'Note that the IASI CO and NH3, and CrIS NH3 are measured at infrared wavelengths and therefore are not sensitive to the smoke plume.' This is not entirely correct. Since smoke is absorbing at mid-IR wavelengths, there should be an influence on the results in case it is not taken into account explicitly in the satellite retrieval procedure. I would be interested if there are any sensitivity calculations for IR retrievals in presence of smoke.*

We have added the following text to Sect. 4.4 to address this:

"At infrared wavelengths, smoke aerosol does not have a strong effect on retrievals. Clarisse et al. (2010a) showed the impact of biomass burning aerosol on IASI retrievals in the 800-1200 cm$^{-1}$ range and found the strongest effect for 1000-1200 cm$^{-1}$.  IASI NH$_3$ is retrieved in the 800-1200 cm$^{-1}$ range, but most of the weight is for 900-980 cm$^{-1}$.  CrIS retrievals are performed for 650-1095 cm$^{-1}$, with the main NH$_3$ absorbing spectral region at 960-970 cm$^{-1}$.  The effect of biomass burning for 900-980 cm$^{-1}$ is small (Clarisse et al., 2010a) and therefore will have minor influence on the IASI and CrIS NH$_3$ retrievals. IASI CO retrievals are performed in the 2128-2206 cm$^{-1}$ wavelength range, where the effect of aerosol is also weaker than in the 1000-1200 cm$^{-1}$ range (Sutherland and Khanna, 1991). Therefore, uncertainties due to the effect of smoke on the VCDs are insignificant compared to other sources of error and were not included in the uncertainty calculations for IASI and CrIS."

*Technical:*

*P13, L32: 'due the' -> 'due to the'*

*P14, L22: 'emissions' -> 'emission'*

*P16, L28: 'then applying conversion factor' -> 'then applying a conversion factor'*

*P18, L5: '1.0' -> '1.0 g/kg'*

*P18, L6: '3.7' -> '3.7 g/kg' and '3.9' -> '3.9 g/kg'*

These have all been corrected as recommended.